# Version 1 of a sea ice module for the physics based, detailed, multi-layer SNOWPACK model

Nander Wever[1,2,3], Leonard Rossmann[4], Nina Maaß[5], Katherine C. Leonard[1,2,6], Lars Kaleschke[4], Marcel Nicolaus[4], and Michael Lehning[2,3]

[1]Department of Atmospheric and Oceanic Sciences, University of Colorado Boulder, Boulder, CO, USA.
[2]CRYOS, School of Architecture, Civil and Environmental Engineering, EPFL, Lausanne, Switzerland.
[3]WSL Institute for Snow and Avalanche Research SLF, Davos, Switzerland.
[4]Alfred Wegener Institute, Helmholtz Centre for Polar and Marine Research, Bremerhaven, Germany.
[5]University of Hamburg, Institute of Oceanography, CEN, Germany.
[6]Cooperative Institute for Research in Environmental Science (CIRES), University of Colorado Boulder, Boulder, CO, USA.

**Correspondence:** NANDER WEVER (nander.wever@colorado.edu)

**Abstract.** Sea ice is an important component of the global climate system. The presence of a snowpack covering sea ice can strongly modify the thermodynamic behaviour of the sea ice, due to the low thermal conductivity and high albedo of snow. The snowpack can be stratified and change properties (density, water content, grain size and shape) throughout the seasons. Melting snow provides fresh water which can form melt ponds, or cause flushing of salt out of the underlying sea ice, while flooding of the snow layer by saline ocean water can strongly impact both the ice mass balance and the freezing point of the snow. To capture the complex dynamics from the snowpack, we introduce modifications to the physics-based, multi-layer SNOWPACK model to simulate the snow – sea ice system. Adaptations to the model thermodynamics and a description of water and salt transport through the snow – sea ice system by coupling the transport equation to the Richards equation were added. These modifications allow the snow microstructure descriptions developed in the SNOWPACK model to be applied to sea ice conditions as well. Here, we drive the model with data from Snow and Ice Mass-balance Buoys installed in the Weddell Sea in Antarctica. The model is able to simulate the temporal evolution of snow density, grain size and shape, and snow wetness. The model simulations show abundant depth hoar layers and melt layers, as well as superimposed ice formation due to flooding and percolation. Gravity drainage of dense brine is underestimated as convective processes are so far neglected. Furthermore, with increasing model complexity, detailed forcing data for the simulations is required, which is difficult to acquire due to limited observations in polar regions.

## 1 Introduction

Sea ice is an important component of the global climate system (e.g., Goosse and Fichefet, 1999; Ferrari et al., 2014). During the freezing process of ocean water, salt is expelled from the ice and dense, saline water is formed. The negative buoyancy of the resulting dense water is an important mechanism driving global ocean circulation (e.g., Gordon, 1988). Sea ice also forms the interface between the ocean and the atmosphere for extended periods of time, altering the surface energy balance (e.g., Ledley, 1991; Brandt et al., 2005; Perovich et al., 2011).

Antarctic sea ice is largely snow covered (Allison et al., 1993), which has major implications for the energy and mass balance of sea ice (Eicken et al., 1995; Massom et al., 2001). The layer of snow strongly modifies the energy balance both via its high albedo (Grenfell and Perovich, 1984; Brandt et al., 2005; Perovich et al., 2011), and through the thermal conductivity of snow, which insulates the sea ice and limits ice growth (Maykut and Untersteiner, 1971; Sturm et al., 2002b). In summer, however, the high albedo of the snow cover may limit ice melt (e.g., Curry et al., 1995). The thermal conductivity of snow varies widely based on the snow microstructural properties, indicating the need to explicitly consider those properties and their spatial and vertical variability (Calonne et al., 2011; Sturm et al., 2002b).

Although the effect of high albedo and low thermal conductivity of snow on the energy balance reduces sea ice growth, snow can also provide a positive contribution to the mass balance of the sea ice. Particularly on Antarctic sea ice, it is regularly observed that the snow cover, due to its weight, pushes the sea ice below sea level. Flooding then may occur via cracks or through the brine channels. Refreezing of the flooded layer is an important mechanism increasing the ice mass in addition to basal thermodynamic growth in the Southern Ocean (Eicken et al., 1994; Jeffries et al., 1997). Snow meltwater or rain percolating and accumulating on top of the ice can also (re)freeze and cause the formation of superimposed ice (Haas et al., 2001; Nicolaus et al., 2003; Obleitner and Lehning, 2004). The spatial and temporal variability of these processes is poorly known due to limited knowledge of snow cover distribution and properties.

Assessing snow amounts on sea ice from atmospheric forcing alone is not straight-forward, as the local precipitation is redistributed by wind, and an unknown fraction of precipitation never accumulates over level ice as it is either blown into leads or accumulates in the lee of surface topography (Déry and Tremblay, 2004; Leonard and Maksym, 2011; Trujillo et al., 2016; Liston et al., 2018; Petty et al., 2018), or is lost due to drifting snow sublimation (e.g., Wever et al., 2009). These processes result in snow depth patterns and accumulation patterns that are typically spatially highly variable (Trujillo et al., 2016; Haas et al., 2017). Also, the wind redistribution and smoothing effect of the snow cover modifies the aerodynamic roughness of the sea ice surface, influencing the momentum flux between atmosphere and sea ice (Andreas, 2002; Weiss et al., 2011) and consequently the large-scale movement of sea ice (Tremblay and Mysak, 1997).

Snow stratigraphy over sea ice can also be highly variable in space and time, and exhibit complex microstructural layering (Massom et al., 1998, 2001; Nicolaus et al., 2009). Strong temperature gradients over shallow snow covers can lead to facetting and grain growth, resulting in layers with facetted depth hoar crystals (Toyota et al., 2007; Lewis et al., 2011). Inhomogeneities in the snow caused by wind slabs, ice layers and melt-freeze crusts have been reported for both Arctic and Antarctic sea ice (e.g., Massom et al., 1997; Sturm et al., 2002a; Gallet et al., 2017; Merkouriadi et al., 2017; Arndt and Paul, 2018). As snow on sea ice is typically up to a few decimeter thick for extended periods of time, and accumulation events are small, it is a challenge for numerical models to capture the vertical variability in the snow cover accurately.

Water transport processes in the snowpack can have a strong impact on the snow microstructure and the salinity distribution in the snow – sea ice system. Surface melt can create downward water percolation, which can refreeze lower down in the snowpack as ice layers or form superimposed ice (Nicolaus et al., 2009), or (particularly in the Arctic) form melt ponds. Upon warming of the underlying sea ice, increased hydraulic conductivity from expanding brine channels may cause freshwater to flush salt out of the underlying ice (e.g., Notz and Worster, 2009). Upward motion of liquid water due to capillary forces has

also been observed (Massom et al., 1998, 2001). Capillary wicking can move salt upward, creating a salty slush layer and frost flowers at the ice surface (Rankin et al., 2002; Domine et al., 2004), which may get buried beneath snow after snow fall, impacting the salinity and water content of the lowest snow layers.

The snow cover is also important for deriving sea ice thickness and snow depth from satellite remote sensing. For example, the microwave signature (e.g., Markus and Cavalieri, 1998; Drinkwater and Crocker, 1988; Powell et al., 2006) and the radar penetration (e.g., Willatt et al., 2010; Ricker et al., 2014) of snow-covered sea ice depend on the properties and layering of the snow.

A wide range of sea ice models have been developed (e.g., Bitz and Lipscomb, 1999; Maksym and Jeffries, 2000; Ukita and Martinson, 2001; Huwald et al., 2005; Griewank and Notz, 2013; Turner and Hunke, 2014). Due to the important role of snow on sea ice, many sea ice models include a description of the snow cover in their models (Lecomte et al., 2011; Notz, 2012). The most important factors considered by sea ice models is the albedo and the insulating effect of the snow. Several studies used the snow thermodynamics model SNTHERM, either using the internal sea ice module or coupled to a sea ice model (Jordan et al., 1999; Nicolaus et al., 2006; Chung et al., 2010; Fuller et al., 2015) to improve the thermodynamical upper boundary condition for sea ice, and assessing the snow microstructure on sea ice.

Rather than improving the representation of snow in an existing sea ice model, this paper presents a new sea ice module developed for the physics-based, detailed, multi-layer snow cover model SNOWPACK. Our motivation is that the SNOWPACK model has a long development history with a focus on accurately representing physical processes in the snow cover. This includes a detailed representation of snow microstructure as well as liquid water flow processes (Bartelt and Lehning, 2002; Lehning et al., 2002a, b; Wever et al., 2015, 2016). The model has previously been successfully applied in Polar regions. For example, for the Greenland Ice Sheet (Steger et al., 2017), it was found to provide accurate simulations of water flow and refreezing processes. An application to the Antarctic Plateau (Groot Zwaaftink et al., 2013) showed good agreement for new snow density and temperature profiles in a drifting snow dominated environment. Here, we apply the new SNOWPACK module to the Antarctic sea ice environment and demonstrate its ability to successfully simulate snow on sea ice conditions.

## 2   Methods

The SNOWPACK model has seen a long development history regarding snow processes. The model calculates the snow energy balance, the resulting temperature distribution, snow settling (densification), liquid water flow and the evolution of snow microstructure. The snow microstructure is described by four parameters: grain size, bond radius, sphericity and dendricity. Grain size is considered the average grain size (Baunach et al., 2001) and bond radius is defined to be the minimum constriction in the connection between to grains (see Fig. 2 in Lehning et al. (2002a)). As defined in Brun et al. (1992), dendricity describes how much of the original crystal shapes are still remaining in a snow layer, where 1 corresponds to perfect dendritic snow, and 0 to perfect rounded or facetted snow. Sphericity describes the ratio of rounded versus angular shapes, where 1 denotes perfectly round shapes, and 0 describes perfectly facetted shapes. Governing state equations describe the time evolution of

these parameters given snow temperature, density, liquid water content (LWC), etc. The full model description is presented in Bartelt and Lehning (2002) and Lehning et al. (2002a, b).

The basic model structure of SNOWPACK is congruent with the mushy-layer theory for sea ice (Hunke et al., 2011; Turner and Hunke, 2014). The model is 1-dimensional, with an arbitrary number of vertical layers of arbitrary depth. Typical layer depth, however, is 1-2 cm. Each layer's total volume is subdivided into a part consisting of respectively ice, water and air. Henceforth these layers are called "snow" layers. Note that SNOWPACK also considers soil as a category, which is hereafter ignored in the context of sea ice.

The sea ice extension of the model provides modifications to the model code to include the effect of salinity on thermal properties and liquid water flow. Furthermore, ice growth and melt at the bottom of the domain is assessed. Flooding is considered to occur only in one-dimension, and lateral advection of liquid water is ignored (Maksym and Jeffries, 2000).

Below, we detail the modifications of the SNOWPACK model to make it suitable for sea ice simulations. The modifications are implemented in the main version of SNOWPACK, and sea ice specific settings are only needed in the configuration files for the model. The code base of the SNOWPACK model is the same and future developments in other parts of the code will also be directly available for the sea ice version.

## 2.1 Heat Equation and Thermodynamics

SNOWPACK solves the heat equation using finite elements, as described in Bartelt and Lehning (2002), allowing to have a simulated snow surface temperature as well as a temperature at the bottom of the snow column. Each snow layer, called element, has two nodes with the adjacent elements. Several modifications were necessary to take into account the effect of salinity on thermodynamical properties.

First, it is assumed that all salt is concentrated in the liquid water in the brine pockets and that the volumetric ice content is free of salt, such that one can write:

$$S_b = \frac{S}{\theta}. \tag{1}$$

Here, $S$ is the bulk salinity (g kg$^{-1}$), $S_{\mathrm{b}}$ is the brine salinity (g kg$^{-1}$) and $\theta$ is the volumetric LWC (m$^3$ m$^{-3}$).

The melting point $T_{\mathrm{m}}$ (K) of each snow element can then be expressed as a function of brine salinity by the commonly used relationship (Assur, 1960):

$$T_{\mathrm{m}} = -\mu S_b + T_0, \tag{2}$$

where $\mu$ is a constant (0.054 K (g kg$^{-1}$)$^{-1}$) and $T_0$ is the melting point of fresh water (273.15 K). Eq. 2 is valid between $0\,°$C and $-6\,°$C (Vancoppenolle et al., 2019).

To solve the heat equation, we assume equilibrium between the element temperature $T_{\mathrm{e}}$ (K) and the brine melting point $T_{\mathrm{m}}$. When the ice is heating (cooling), brine volume is assumed to increase (decrease) instantaneously by phase changes in the surrounding ice, in order to maintain $T_{\mathrm{e}} = T_{\mathrm{m}}$. This is achieved by feeding back the energy associated with the phase change as a source/sink term in the heat equation (see Eq. 3 in Bartelt and Lehning (2002)). Note that the latent heat released when

water freezes increases the temperature locally, and vice versa. These effects on local temperature reduce the energy released during freezing or energy available for melt. Additionally, the refreezing or melting of ice impacts brine salinity and thereby the melting temperature. These competing processes slow down the convergence in the solver for the heat equation, which can be mitigated by providing an improved estimate of the melting temperature which satisfies: (i) the condition provided by Eq. 1 for the new LWC, (ii) Eq. 2 for the new melting temperature given the new brine salinity, and (iii) the change in ice content for the given deviation of the new element temperature from the melting point, by algebraically solving these three conditions with the three unknowns to find the new melting temperature.

The bottom node of the domain represents the interface between sea ice and ocean and its temperature is prescribed as a Dirichlet boundary condition using the freezing temperature of the ocean water, which is determined by the prescribed ocean salinity (set to 35 g kg$^{-1}$ in this study). Typically heat is advected into the sea ice from the ocean below, referred to as the ocean heat flux. We determine the net energy loss or gain at the bottom node, given the prescribed ocean heat flux and the internal heat flux in the lowest sea ice element. This net energy is translated into bottom ice growth or melt.

An uncertainty for ice growth is the ice porosity of the newly formed ice. We apply a similar approach to the one presented in Griewank and Notz (2013), where an ice content threshold is defined. When the lowest element has an ice content above this value, the net energy is used to create new ice elements with a brine salinity equal to ocean salinity (Vancoppenolle et al., 2010). Otherwise, the bottom element grows in length, after first increasing the ice content to the threshold. We set a threshold of 0.99 m$^3$ m$^{-3}$, which we also prescribe as the maximum allowed ice content of a single layer. Mass loss is applied by reducing the element length.

## 2.2 Brine Dynamics

The SNOWPACK model is equipped with a solver for the full Richards equation for transport in porous media (Wever et al., 2014). Here, we modified the solver for the Richards equation to account for density variations and couple the Richards equation to a transport equation for salinity. This provides an explicit treatment of brine dynamics. First, the Richards equation solves the liquid water flow in the snow – sea ice system, keeping the salinity constant with time. After each integration of the Richards equation, the advection-diffusion equation is solved for the same time step under the assumption of constant liquid water fluxes. The time step is limited to a maximum of 15 minutes, although stability criteria for both the Richards equation as well as the advection-diffusion equation for salinity may impose additional, stricter, time constraints. Below, we detail how this scheme for liquid water transport was modified for sea ice.

### 2.2.1 Richards Equation for Water Flow

The mixed form of the Richards equation reads:

$$\frac{\partial \theta}{\partial t} - \frac{\partial}{\partial z}\left[\frac{\kappa}{\eta}\frac{\partial p}{\partial z}\right] - s = 0, \tag{3}$$

where $t$ is time (s), $z$ is the vertical coordinate (m), $\kappa$ is the permeability (m$^2$), $\eta$ is the viscosity (m s$^{-2}$) and $s$ is a source/sink term (m$^3$ m$^{-3}$ s$^{-1}$). The pressure $p$ can be considered the sum of water potential and gravity potential:

$$p = \rho g h + \rho g z \cos(\gamma) \tag{4}$$

where $h$ is the pressure head (m), $g$ is the gravitational acceleration (m s$^{-2}$), $\rho$ the density of the flowing liquid (kg m$^{-3}$) and

5 $\quad$ $\gamma$ is the slope angle. Because SNOWPACK can be used in sloped terrain, we keep this term for completeness of the model description, although $\gamma$ is obviously 0 for sea ice.

The density of liquid water ($\rho$) is adjusted for salinity according to:

$$\rho = \rho_{\mathrm{w}} + \beta S_{\mathrm{b}}, \tag{5}$$

where $\rho_{\mathrm{w}}$ is the density of fresh liquid water (1000 kg m$^{-3}$), $\beta$ is a salinity coefficient, approximated as 0.824 kg$^2$ g$^{-1}$ m$^{-3}$

$\quad$ (Massel, 2015, Appendix A).

The permeability $\kappa$ is replaced by the hydraulic conductivity $K$, which relates to $\kappa$ via:

$$K = \kappa \frac{\rho g}{\mu}, \tag{6}$$

where $\mu$ is the dynamic viscosity of water (0.001792 kg (m s)$^{-1}$).

A critical assumption in the typical application of the Richards equation is that both $g$ and $\rho$ are constant in time and $z$, and

15 $\quad$ consequently can be eliminated from the equation. Due to salinity variations in sea ice, variations of density of the flowing liquid can occur and are actually considered the driving mechanism in the temporal and spatial evolution of the salinity of sea ice.

Therefore, we rewrite the Richards equation for sea ice by considering $\rho$ as a function of $z$ and only eliminating $g$, arriving at:

$\quad$ $$\frac{\partial \theta}{\partial t} - \frac{\partial}{\partial z}\left[\frac{K}{\rho}\frac{\partial}{\partial z}\left(\rho h + \rho z \cos(\gamma)\right)\right] - s = 0, \tag{7}$$

As outlined in Wever et al. (2014), the Richards equation is implemented in SNOWPACK by the discretization proposed by Celia et al. (1990). The backward Euler approximation in time coupled with a simple Picard iteration, as shown in Eq. 14 of Celia et al. (1990), becomes, for Eq. 7:

$$\frac{\theta^{n+1,m+1} - \theta^n}{\Delta t} - \frac{\partial}{\partial z}\left(\frac{K^{n+1,m}}{\rho}\frac{\partial \rho h^{n+1,m+1}}{\partial z}\right) - \cos(\gamma)\frac{\partial K^{n+1,m}}{\partial z} - \cos(\gamma)\frac{\partial}{\partial z}\left(\frac{K^{n+1,m} z}{\rho}\frac{\partial \rho}{\partial z}\right) - s = 0, \tag{8}$$

$\quad$ where $n$ and $m$ denote the time and iteration level, respectively. Here, we have used the chain rule to write:

$$\frac{\partial}{\partial z}\left[\frac{K}{\rho}\frac{\partial}{\partial z}(\rho z)\right] = \frac{\partial}{\partial z}\left[K\frac{\partial z}{\partial z}\right] + \frac{\partial}{\partial z}\left[\frac{K}{\rho}z\frac{\partial \rho}{\partial z}\right] = \frac{\partial K}{\partial z} + \frac{\partial}{\partial z}\left[\frac{K}{\rho}z\frac{\partial \rho}{\partial z}\right] \tag{9}$$

The last term on the right hand side expresses the liquid water flow driven by density differences.

After applying a Taylor expansion to Eq. 15 of Celia et al. (1990), and defining $\delta^m \equiv \rho h^{n+1,m+1} - \rho h^{n+1,m}$, we arrive at:

$$\left(\frac{1}{\Delta t}\frac{C^{n+1,m}}{\rho}\right)\delta^m + \frac{\theta^{n+1,m}-\theta^n}{\Delta t} - \frac{\partial}{\partial z}\left(\frac{K^{n+1,m}}{\rho}\frac{\partial \rho h^{n+1,m+1}}{\partial z}\right) - \cos(\gamma)\frac{\partial K^{n+1,m}}{\partial z} - \cos(\gamma)\frac{\partial}{\partial z}\left(\frac{K^{n+1,m}z}{\rho}\frac{\partial \rho}{\partial z}\right) - s = 0.$$
(10)

Finally, Eq. 16 in Celia et al. (1990) becomes:

$$\left(\frac{1}{\Delta t}\frac{C^{n+1,m}}{\rho}\right)\delta^m - \frac{\partial}{\partial z}\left(\frac{K^{n+1,m}}{\rho}\frac{\partial \delta^m}{\partial z}\right)$$

$$= \frac{\partial}{\partial z}\left(\frac{K^{n+1,m}}{\rho}\frac{\partial \rho h^{n+1,m+1}}{\partial z}\right) + \cos(\gamma)\frac{\partial K^{n+1,m}}{\partial z} + \cos(\gamma)\frac{\partial}{\partial z}\left(\frac{K^{n+1,m}z}{\rho}\frac{\partial \rho}{\partial z}\right) - \frac{\theta^{n+1,m}-\theta^n}{\Delta t} + s \quad (11)$$

After applying the standard finite difference approximation in space, Eq. 16 in Celia et al. (1990) translates into ($i$ denoting the spatial coordinate):

$$\frac{C_i^{n+1,m}}{\rho_i}\frac{\delta_i^m}{\Delta t} - \frac{1}{(\Delta z)^2}\left[\frac{K_{i+1/2}^{n+1,m}}{\rho_{i+1/2}}\left(\delta_{i+1}^m - \delta_i^m\right) - \frac{K_{i-1/2}^{n+1,m}}{\rho_{i-1/2}}\left(\delta_i^m - \delta_{i-1}^m\right)\right]$$

$$= \frac{1}{(\Delta z)^2}\left[\frac{K_{i+1/2}^{n+1,m}}{\rho_{i+1/2}}\left(\rho_{i+1}h_{i+1}^{n+1,m} - \rho_i h_i^{n+1,m}\right) - \frac{K_{i-1/2}^{n+1,m}}{\rho_{i-1/2}}\left(\rho_i h_i^{n+1,m} - \rho_{i-1}h_{i-1}^{n+1,m}\right)\right]$$

$$+ \cos(\gamma)\frac{K_{i+1/2}^{n+1,m} - K_{i-1/2}^{n+1,m}}{\Delta z} + \cos(\gamma)\frac{\frac{K_{i+1/2}^{n+1,m}}{\rho_{i+1/2}}z_{i+1/2}\frac{\rho_{i+1}-\rho_i}{\Delta z} - \frac{K_{i-1/2}^{n+1,m}}{\rho_{i-1/2}}z_{i-1/2}\frac{\rho_i-\rho_{i-1}}{\Delta z}}{\Delta z} - \frac{\theta_i^{n+1,m}-\theta_i^n}{\Delta t} + s_i$$

$$\equiv \left(R_i^{n+1,m}\right)_{\mathrm{MPFD}}, \quad (12)$$

where the notation $R_{\mathrm{MPFD}}$ is kept for consistency with Celia et al. (1990). The system of equations described by Equation 12 forms a tri-diagonal matrix. As in Wever et al. (2014), the function `DGTSV` from the `LAPACK` library (Anderson et al., 1999) is called to compute the solution. When `LAPACK` is not available, or not selected on compile time, the Thomas algorithm is used as the implemented default alternative, which does not depend on external libraries. However the Thomas algorithm is not the preferred option as in contrast to `DGTSV`, it lacks partial pivoting and may suffer from numerical instabilities.

### 2.2.2 Transport Equation for Salinity

The governing equation in 1-dimension for concentration describes the change in salinity as a combination of a diffusion and advection process:

$$\frac{\partial}{\partial t}(\theta S_{\mathrm{b}}) - \frac{\partial}{\partial z}\left(D\theta\frac{\partial S_{\mathrm{b}}}{\partial z}\right) - \frac{\partial}{\partial z}(qS_{\mathrm{b}}) - s_{\mathrm{sb}} = 0 \qquad (13)$$

Where $D$ is the diffusion coefficient (m$^2$ s$^{-1}$), considered here, as a first approximation, independent of temperature. In this study, $D$ is set as $10^{-10}$ m$^2$ s$^{-1}$ (Poisson and Papaud, 1983). $q$ denotes the liquid water flux (m s$^{-1}$) and $s_{\mathrm{sb}}$ is a source/sink term for salinity (assumed here to be 0 g kg$^{-1}$ s$^{-1}$).

An implicit numerical scheme for Equation 13 becomes in discretized form ($n$ and $i$ again denoting time and spatial level, respectively):

$$\frac{\left(\theta_i^{n+1} S_{\mathrm{b},i}^{n+1} - \theta_i^n S_{\mathrm{b},i}^n\right)}{\Delta t}$$

$$-f\left[\left(\frac{2D_{i+1}^n \theta_{i+1}^{n+1} S_{\mathrm{b},i+1}^{n+1}}{\Delta z_{\mathrm{up}}\left(\Delta z_{\mathrm{up}} + \Delta z_{\mathrm{down}}\right)} - \frac{2D_i^n \theta_i^{n+1} S_{\mathrm{b},i}^{n+1}}{\left(\Delta z_{\mathrm{up}} \Delta z_{\mathrm{down}}\right)} + \frac{2D_{i-1}^n \theta_{i-1}^{n+1} S_{\mathrm{b},i-1}^{n+1}}{\Delta z_{\mathrm{down}}\left(\Delta z_{\mathrm{up}} + \Delta z_{\mathrm{down}}\right)}\right)\right]$$

$$-(1-f)\left[\left(\frac{2D_{i+1}^n \theta_{i+1}^n S_{\mathrm{b},i+1}^n}{\Delta z_{\mathrm{up}}\left(\Delta z_{\mathrm{up}} + \Delta z_{\mathrm{down}}\right)} - \frac{2D_i^n \theta_i^n S_{\mathrm{b},i}^n}{\left(\Delta z_{\mathrm{up}} \Delta z_{\mathrm{down}}\right)} + \frac{2D_{i-1}^n \theta_{i-1}^n S_{\mathrm{b},i-1}^n}{\Delta z_{\mathrm{down}}\left(\Delta z_{\mathrm{up}} + \Delta z_{\mathrm{down}}\right)}\right)\right]$$

$$-f\left[\left(\frac{q_{i+1}^n S_{\mathrm{b},i+1}^{n+1} - q_{i-1}^n S_{\mathrm{b},i-1}^{n+1}}{\left(\Delta z_{\mathrm{up}} + \Delta z_{\mathrm{down}}\right)}\right)\right] - (1-f)\left[\left(\frac{q_{i+1}^n S_{\mathrm{b},i+1}^n - q_{i-1}^n S_{\mathrm{b},i-1}^n}{\left(\Delta z_{\mathrm{up}} + \Delta z_{\mathrm{down}}\right)}\right)\right] - s_{\mathrm{sb}} = 0 \quad (14)$$

Here, taking $f = 1$ results in a fully implicit scheme, whereas taking $f = 0.5$ corresponds to the Crank-Nicolson scheme (Crank and Nicolson, 1996). The equation is solved after every time step for liquid water flow. Then, for LWC $\theta$, both $\theta^n$, as well as $\theta^{n+1}$ are known from solving Eq. 12. Furthermore, it is assumed that the water flux $q$ is constant with time and can be referenced with the time level $n$. The liquid water flux is directly obtained from the Darcy-Buckingham law (Buckingham, 1907), which forms the basis of Richards equation:

$$q_{i+1}^n = \frac{K_{i+1/2}}{\rho_{i+1/2}}\left(\frac{\rho_{i+1} h_{i+1} - \rho_i h_i}{\left(\Delta z_{\mathrm{up}}\right)^2}\right) + \cos\left(\gamma\right) K_{i+1/2} + \cos\left(\gamma\right)\frac{K_{i+1/2}}{\rho_{i+1/2}} z_{i+1/2}\left(\frac{\rho_{i+1} - \rho_i}{\Delta z_{\mathrm{up}}}\right), \quad (15)$$

and

$$q_{i-1}^n = \frac{K_{i-1/2}}{\rho_{i-1/2}}\left(\frac{\rho_i h_i - \rho_{i-1} h_{i-1}}{\left(\Delta z_{\mathrm{down}}\right)^2}\right) + \cos\left(\gamma\right) K_{i-1/2} + \cos\left(\gamma\right)\frac{K_{i-1/2}}{\rho_{i-1/2}} z_{i-1/2}\left(\frac{\rho_i - \rho_{i-1}}{\Delta z_{\mathrm{down}}}\right). \quad (16)$$

The domain in SNOWPACK is typically nonuniform and the spatial discretizations in Eq. 14 for a nonuniform grid are based on Veldman and Rinzema (1992). The system of equations described by Equation 14 forms a tri-diagonal matrix, similar to (and solved in the same way) as the equation for liquid water flow (Eq. 12).

SNOWPACK by default uses the Crank-Nicolson scheme. The fully implicit scheme is only first order accurate, whereas the Crank-Nicolson scheme is second order accurate. Both schemes are unconditionally stable and suffer only minimal numerical diffusion for advection. As with many other common schemes, the advection part does not perfectly conserve sharp transitions. The Crank Nicolson scheme is, in spite of being unconditionally stable, prone to spurious oscillations in the solution. To choose adequate time steps, we apply the Courant-Friedrichs-Lewy (CFL) condition (Courant et al., 1928), typically required for stability of an explicit scheme, also for the Crank-Nicolson scheme:

$$q\frac{\Delta t}{\Delta z} \leq 1 \quad (17)$$

Note that if the CFL condition is violated, the time step is reduced and the last time step for the Richards equation is also repeated with the reduced time step.

### 2.2.3 Boundary Conditions

The boundary conditions for the Richards equation (Eq. 3) are determined by a Neumann boundary condition at the top, consisting of the top water flux from rain, evaporation or condensation and a Dirichlet boundary condition at the bottom by prescribing the pressure head. The pressure head at the bottom of the sea ice corresponds to the water pressure at that depth, which equals the sea level in the model domain ($z_{sl}$, m). This is determined from the isostatic balance:

$$z_{sl} = \frac{\text{SWE}}{(\rho_w + \beta S_o)}, \tag{18}$$

where $S_o$ is the ocean water salinity (g kg$^{-1}$) and SWE is the snow water equivalent, defined as the sum over all elements $N$ of the mass of each element $j$ in the model:

$$\text{SWE} = \sum_{j=1}^{N} (\theta_{i,j}\rho_i + \theta_j\rho_j)\Delta z_j, \tag{19}$$

where $\theta_{i,j}$ and $\theta_j$ are the volumetric content (m$^3$ m$^{-3}$) for layer $j$ of ice and water, respectively. $\rho_i$ is the density of freshwater ice (917 kg m$^{-3}$) and $\rho_j$ the brine density of layer $j$ (see Eq. 5).

The boundary conditions for the advection terms of the advection-diffusion equation (Eq. 13) are prescribed as a Neumann boundary condition, with fresh water at the top, and ocean salinity at the bottom. This means that at the bottom, a downward (outgoing) flux removes salt from the domain, and an upward (incoming) flux carries ocean salinity. A downward (incoming) flux at the top of the domain is assumed to be a fresh water flux. Only in case of an outgoing water flux at the top of the domain (evaporation), a zero-flux condition for salinity is used (i.e., salt will remain in the sea ice or snow with evaporation).

For the diffusion terms in Eq. 13, we implemented a no-flux boundary condition at the top of the domain. This implies that there is no diffusion of salt into the atmosphere. This boundary condition is derived by considering the central differences scheme for the diffusion term on a nonuniform grid, determined according to:

$$\frac{\partial^2}{\partial z^2}(D\theta S_b) \approx \frac{\frac{\partial D\theta S_b}{\partial z}\big|_{z_{i+1/2}} - \frac{\partial D\theta S_b}{\partial z}\big|_{z_{i-1/2}}}{z_{i+1/2} - z_{i-1/2}} \approx \frac{\frac{D_{i+1}\theta_{i+1}S_{b,i+1} - D_i\theta_i S_{b,i}}{z_{up}} - \frac{D_i\theta_i S_{b,i} - D_{i-1}\theta_{i-1}S_{b,i-1}}{z_{down}}}{\frac{1}{2}(z_{up} + z_{down})} \tag{20}$$

In general, a no-flux boundary condition can be achieved by forcing gradients over the boundaries to be 0, such that either the left (upper boundary) or right (lower boundary) term in the numerator of Eq. 20 vanishes.

We only apply the no-flux boundary at the top. For the bottom boundary condition at $i = 0$, diffusion is calculated by assuming:

$$S_{b,i}^n = S_{b,i}^{n+1} = S_o. \tag{21}$$

### 2.2.4 Hydraulic Properties

For solving the Richards equation and the salinity transport equation, several parameters which depend on the snow and ice microstructure need to be specified. For saturated hydraulic conductivity ($K_{\mathrm{sat}}$), we define elements with a porosity (i.e., $1-\theta_{\mathrm{i}}$) larger than 0.25 as snow, and smaller than 0.25 as ice.

5      For snow elements, a formulation based on Calonne et al. (2012) is typically used (see Wever et al. (2014)):

$$K_{\mathrm{sat}} = \left(\frac{\rho g}{\mu}\right) \left[3.0 \left(\frac{r_{\mathrm{es}}}{1000}\right)^2 \exp\left(-0.013\theta_{\mathrm{i}}\rho_{\mathrm{i}}\right)\right],  \tag{22}$$

where $r_{\mathrm{es}}$ is the equivalent sphere radius (m). Note that Eq. 14 in Wever et al. (2014) erroneously shows a factor $0.75$ (which corresponds to $r_{\mathrm{es}}$ being a grain diameter) instead of $3.0$.

     For sea ice, the saturated hydraulic conductivity ($K_{\mathrm{sat}}$) is based on Golden et al. (2007):

$$10 \quad K_{\mathrm{sat}} = 3 \cdot 10^{-8} \left(\frac{\rho g}{\mu}\right) (1-\theta_{\mathrm{i}})^3  \tag{23}$$

     In unsaturated conditions, the van Genuchten-Mualem model (Mualem, 1976) is used to relate the hydraulic conductivity in saturated conditions (Eq. 22 and 23) to unsaturated conditions. For averaging the hydraulic conductivity between elements, we use the geometric mean (Wever et al., 2015), which is the preferred method (Haverkamp and Vauclin, 1979; Celia et al., 1990). Furthermore, in unsaturated conditions, the van Genuchten model is used for the water retention curve, which describes
15   the relationship between capillary suction and LWC (van Genuchten, 1980). The coefficients in this parameterization of water retention in snow is based on the work by Yamaguchi et al. (2012). We extend this parameterization over the full porosity range, in absence of any information of water retention in sea ice as a function of salinity and LWC. It has a relatively small impact, as the largest part of the sea ice is below sea level and the model typically simulates saturated conditions here.

## 3 Data and Simulation Setup

### 20   3.1 In-situ Buoys

We apply the sea ice version of SNOWPACK to snow and sea ice properties measured from two Snow Buoys (Nicolaus et al., 2017) and one Ice Mass-balance Buoy (IMB) in the Weddell Sea, Antarctica. Snow Buoys are autonomous ice tethered instruments, which measure snow surface changes/accumulation with four ultrasonic sensors at approximately 1.5 m above the snow – ice interface (Nicolaus et al., in prep). The four sensors were used to identify if the buoys remained stable on the ice.
25   We construct a time series of the surface elevation referenced to the initial snow – ice interface upon installation of the buoy by averaging the four ultrasonic sensors. In addition to the surface elevation, the Snow Buoys measure barometric air pressure and air temperature.

     Each IMB consists of a 4.8 m long thermistor chain, with a vertical sensor spacing of 0.02 m, and measures sea ice temperatures. From the measurements, interfaces between air, snow, sea ice, and sea water can be determined. The instruments are

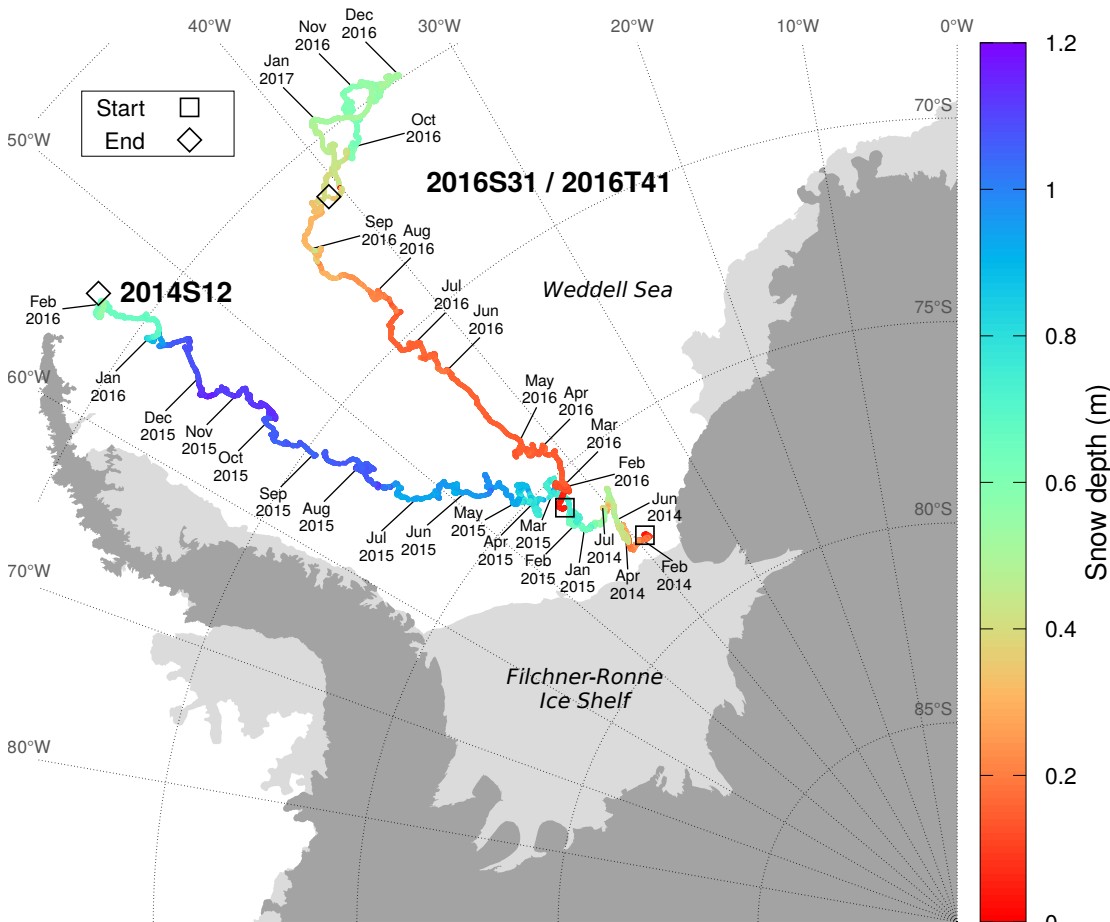

**Figure 1.** Trajectories of the two snow buoys used for the simulations. The average snow depth measured by the four snow depth sensors from each Snow Buoy is shown in color. Labels show the position of the respective Snow Buoy on the 1st day of the month. For buoy 2014S12, some labels at the beginning have been omitted for readability. The deployment location (start) is denoted by a square, the location of the last received data from the buoy (end) is denoted by a diamond. The collocated IMB buoy for the 2014S12 Snow Buoy (2014T9) stopped transmitting data shortly after installation and is ignored in the analysis. The collocated IMB buoy to Snow Buoy 2016S31 is 2016T41.

described by Jackson et al. (2013). It turned out to be good practice to co-deploy Snow Buoys and thermistor chain IMBs in order to observe snow depth and sea ice properties at the same time. The full data set of all these Lagrangian observations is available from http://www.meereisportal.de (Grosfeld et al., 2015).

The two selected Snow Buoys (Snow Buoys 2014S12 (Nicolaus and Schwegmann, 2017) and 2016S31 (Arndt et al., 2017))
5 have long time series and cover different trajectories, as shown in Fig. 1. Unfortunately, IMB 2014T9 collocated with Snow Buoy 2014S12 stopped transmitting data soon after deployment. IMB 2016T41, collocated with Snow Buoy 2016S31 measured

for almost the same period as the Snow Buoy. However, comparisons are limited to the sea ice temperature and ice thickness, excluding the snow cover on top, because the IMB was deployed directly onto the snow surface, without room to accommodate for snow accumulations after installation. Thus, the thermistor chain does not measure the snow cover properties, except for the lowest few cm.

Fig. 1 shows the trajectories with labels marking the location at the 1st of each month. Snow Buoy 2014S12 was deployed on 17 Jan 2014 and remained in the same area very close to the Filchner-Ronne Ice Shelf for the first year. From February 2015 onward, the Snow Buoy drifted northward parallel to the Antarctic Peninsula until data transmission was lost on 1 Feb 2016. During the last 18 hours no valid snow depth data were transmitted and the drift speeds were relatively high, suggesting that it is not a transmission or data logger failure, but rather an indication that flow deformation or breakup is the likely cause of the loss of the Snow Buoy. We consider this particular Snow Buoy due to its long time span, even though a collocated IMB data set is not available.

Snow Buoy 2016S31, collocated with IMB Buoy 2016T41, was deployed on 16 Jan 2016. This deployment first drifted on a predominantly northward course. Around the 1st of December, the northernmost position was reached and the deployment drifted southward again. The Snow Buoy transmitted data until 15 Jan 2017, 2:00 UTC, shortly before the last data transmission by the IMB Buoy (5 Feb 2017, 07:13 UTC). This combination of Snow Buoy and IMB Buoy is interesting for the long time span of collocated measurements.

## 3.2  Initial Conditions

To start each simulation, a description of the initial sea ice state is required. Upon installation of each Snow Buoy, the ice thickness, snow thickness and freeboard were determined. For simulations of these Snow Buoys, we distinguish three categories: (i) sea ice below sea level (ice thickness minus freeboard), (ii) sea ice above sea level (freeboard) and (iii) snow.

For the part of the sea ice below sea level, the volumetric ice content $\theta_i$ was fixed to $0.95$, and the volumetric water content $\theta_w$ was subsequently calculated as:

$$\theta_w = (1 - \theta_i)\frac{\rho_i}{\rho_w} \tag{24}$$

This formulation leaves a small volumetric air content which can be filled when water refreezes and thereby expands. This is currently required for the stability of the numerical schemes in the SNOWPACK model, but in reality refreezing water would increase the pressure in the brine. The element temperature was initialized by the value recorded by the IMB upon installation. As it takes time for the thermistor chain to freeze into the ice and adapt to the surrounding ice temperature, this temperature is mostly representative of the ocean water. The brine salinity was set as the salinity for which the melting point corresponds to the measured temperatures.

For the part of sea ice above sea level, the volumetric ice content $\theta_i$ was also fixed to $0.95$, but the remaining space was assigned to air content and the bulk salinity was set to $0$ g kg$^{-1}$. In field studies, it was found that brine may exist above sea level, due to capillary wicking, or brine content is retained because of low conductivity in the brine channels (e.g., Cox and Weeks, 1974; Massom et al., 2001).

The initial snow layer was 10 cm for Snow Buoy 2014S12 and 2 cm for 2016S31. The snow cover was initialized with a density of 275 kg m$^{-3}$, and a grain size of 0.15 mm. The grain shape was initialized as depth hoar with a sphericity and dendricity of 0. As the majority of the snowpack builds during the model simulations, the simulations are rather insensitive to the choice of initial snowpack properties. The element temperature was derived from the thermistor chain measurements.

5    Finally, the depth of each element was set to 0.02 m.

## 3.3    Forcing Data

Simulations with the SNOWPACK model require air temperature, relative humidity, incoming shortwave radiation, incoming longwave radiation, wind speed and precipitation. Here, we used the European Centre for Medium-range Weather Forecasts ECMFW Reanalysis 5 (ERA5, ERA, 2017) data to provide these parameters to drive the simulations. For each timestep and

10    location, the simulated weather at the closest grid point in the ERA5 model was taken.

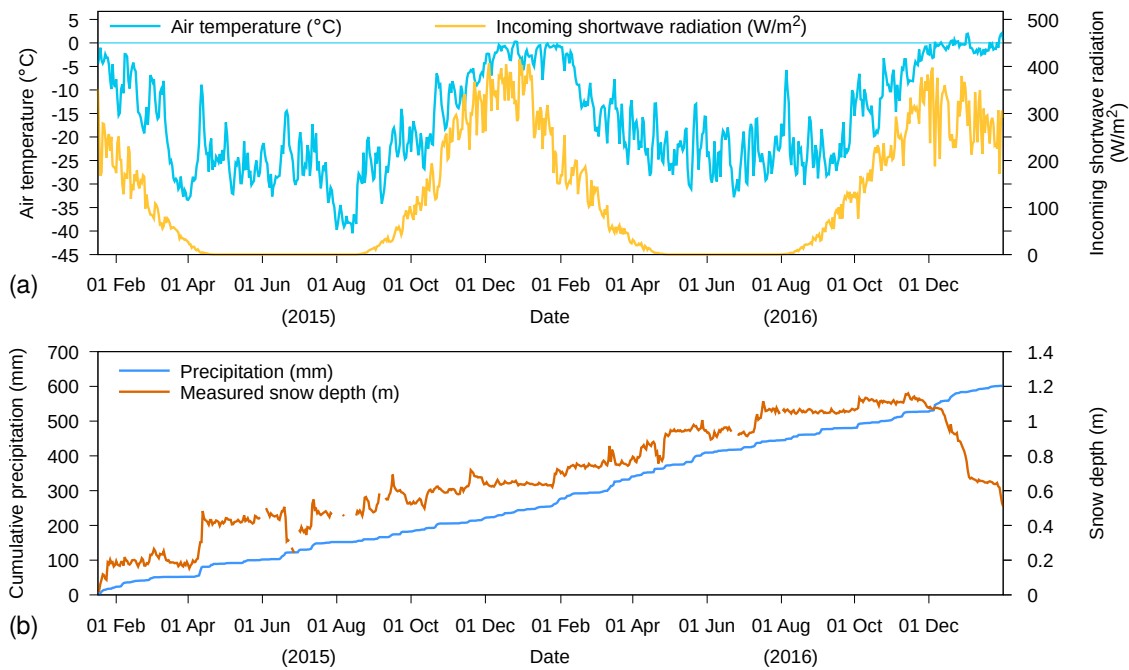

**Figure 2.** Meteorological forcing data from ERA5 for Snow Buoy 2014S12 for (a) daily average air temperature and daily average incoming shortwave radiation, and (b) cumulative precipitation. In (b), also measured snow depth by the buoy is shown.

Fig. 2 shows the daily average air temperature, incoming shortwave radiation and cumulative precipitation from the ERA5 forcing for Snow Buoy 2014S12. Additionally, the measured snow depth by the buoy is shown. We find that the daily average air temperature was mostly below 0 °C, reaching −40 °C in 2015. Near the end of the time series, when the Snow Buoy stopped transmitting data, daily average air temperature varied around 0 °C. It can be expected that positive temperatures during midday is associated with enhanced snow melt, which is indicated by the rapid decrease of measured snow depth

starting from December 2016 onward. During austral winter 2015 and 2016, the Snow Buoy was located below the polar circle and consequently, there was no incoming shortwave radiation. In austral summer 2016/2017, the Snow Buoy drifted above the polar circle. Nevertheless, the average incoming shortwave radiation in the first austral summer is similar to the second austral summer. The cumulative precipitation from ERA5 reached around 600 mm for the almost two years that the buoy

was operative. The buoy recorded 1.2 m of snow accumulation. A marked increase in snow depth around April 1, 2015 is accompanied by only a relatively small precipitation event. Except for this event, both the snow depth as well as the cumulative precipitation show a gradual increase over the two years until the melt phase starts.

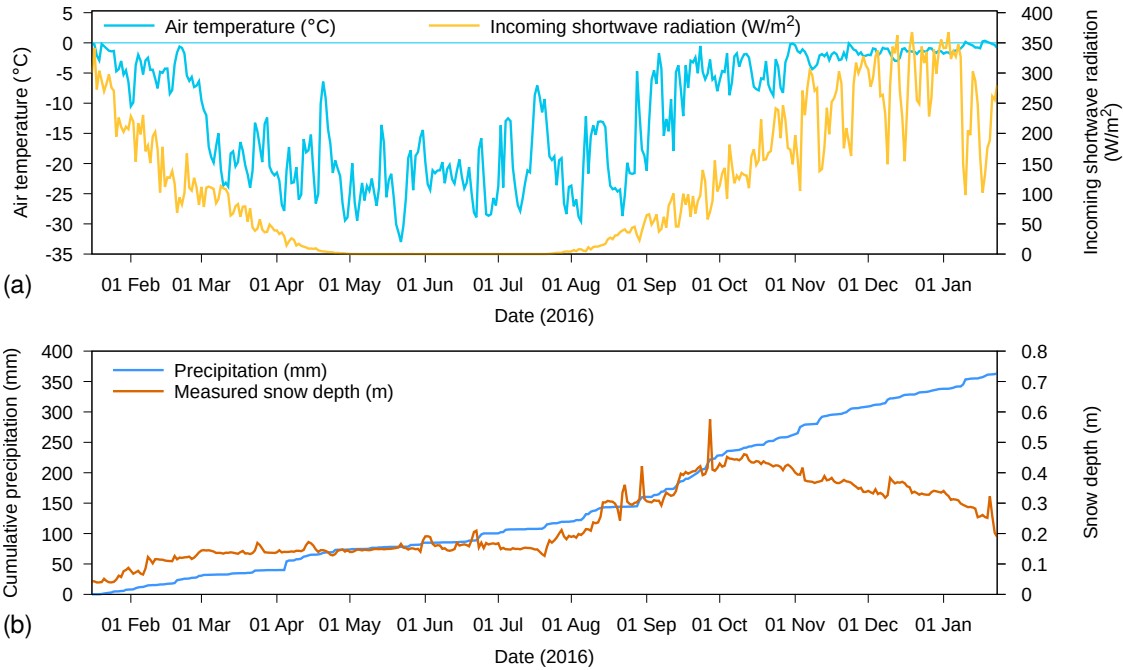

**Figure 3.** Meteorological forcing data from ERA5 for Snow Buoy 2016S31 for (a) daily average air temperature and daily average incoming shortwave radiation, and (b) cumulative precipitation. In (b), also measured snow depth by the buoy is shown.

Fig. 3 shows the daily average air temperature, incoming shortwave radiation and cumulative precipitation from the ERA5 forcing for Snow Buoy 2016S31. The yearly cycle in air temperature is similar to the year 2016 for Snow Buoy 2014S12,

with daily air temperatures reaching around $-30\,°C$ in austral winter. In austral summer, air temperatures around $0\,°C$ suggest melting conditions, particularly from November 2016 to January 2017. This is reflected by the decrease in measured snow depth for this period. The Snow Buoy was also located south of the polar circle, resulting in the absence of shortwave radiation during austral winter. The precipitation sum for the year that the Snow Buoy was operative amounted to 350 mm. The snow depth does not change between March and July 2016, and shows an increase between August and October, after which a

decrease occurs. The cumulative precipitation show a similar pattern, with low precipitation between March and July, followed

by a steady increase afterwards. Only during the melt phase, the increase in cumulative precipitation is not reflected by an increase in snow depth.

SNOWPACK has the possibility to either use a precipitation time series as input to determine when snow fall occurs, or to use a time series of snow depth to interpret increases in measured snow depth as snow fall events when simulated snow depth is less than the measured snow depth (Lehning et al., 1999). Note that there is no downward correction when the measured snow depth is below the simulated snow depth (in case of underestimated melt, surface sublimation or settling, or snow erosion by wind, etc.). In order to use the snow depth driven method for the Snow Buoys and base the mass balance on Snow Buoy data, a layer can be marked in the simulations and tracked throughout the snow – sea ice continuum. By marking the layer that corresponds to the reference level for the snow depth measurements, the measured snow depth can be tracked relative to this marked layer. The output routines of the model have been adapted accordingly to reference the output to either the sea level, or to the marked reference layer.

The ocean heat flux determines the ice mass balance at the bottom of the ice. Its value can be highly variable and dependent on ocean conditions below the sea ice (Ackley et al., 2015). From that study, we use a value of 8 W m$^{-2}$, unless otherwise noted.

## 4 Results

### 4.1 Example Simulation

Fig. 4 shows an example of model behaviour when an initially dry and fresh ice layer with a thickness of 1.58 m, consisting of $94\%$ ice and $6\%$ air expressed as volumetric content, is positioned in ocean water with a salinity of 35 g kg$^{-1}$. The positive pressure head at the bottom of the sea ice corresponds with the pressure exerted by the displaced water. As a consequence, saline water enters the ice matrix (Fig. 4a,c), until it is in equilibrium with the sea level. The initial rate follows the saturated hydraulic conductivity for sea ice with a pore space of $6\%$, which is $3.55 \times 10^{-5}$ m s$^{-1}$ (Eq. 23).

As the pressure difference of the liquid water inside the ice matrix and the surrounding ocean water is decreasing, the salt influx rate decreases over time (Fig. 4e). The brine salinity corresponds to the salinity of ocean water (35 g kg$^{-1}$, see Fig. 4d), corresponding to a bulk salinity of 1.65 g kg$^{-1}$ (see Fig. 4c). The added mass to the sea ice (Fig. 4b) causes the ice to sink deeper inside the ocean water, decreasing the freeboard (Fig. 4f).

This example illustrates that the Crank-Nicolson scheme does not preserve the sharp transition in salinity, as both the bulk (Fig. 4c) as well as the brine (Fig. 4d) salinity shows smoothing behaviour at the saline water front. This reduces the pore space by refreezing to maintain thermal equilibrium, reducing the saturated hydraulic conductivity in the wetting front region.

### 4.2 Temperature Validation

Fig. 5a and 5b show the simulated temperature of the snow–sea ice system for simulations driven by in-situ measured snow depth and ERA5 precipitation, respectively. Dashed lines denote the reference level, i.e., the snow – ice interface as determined

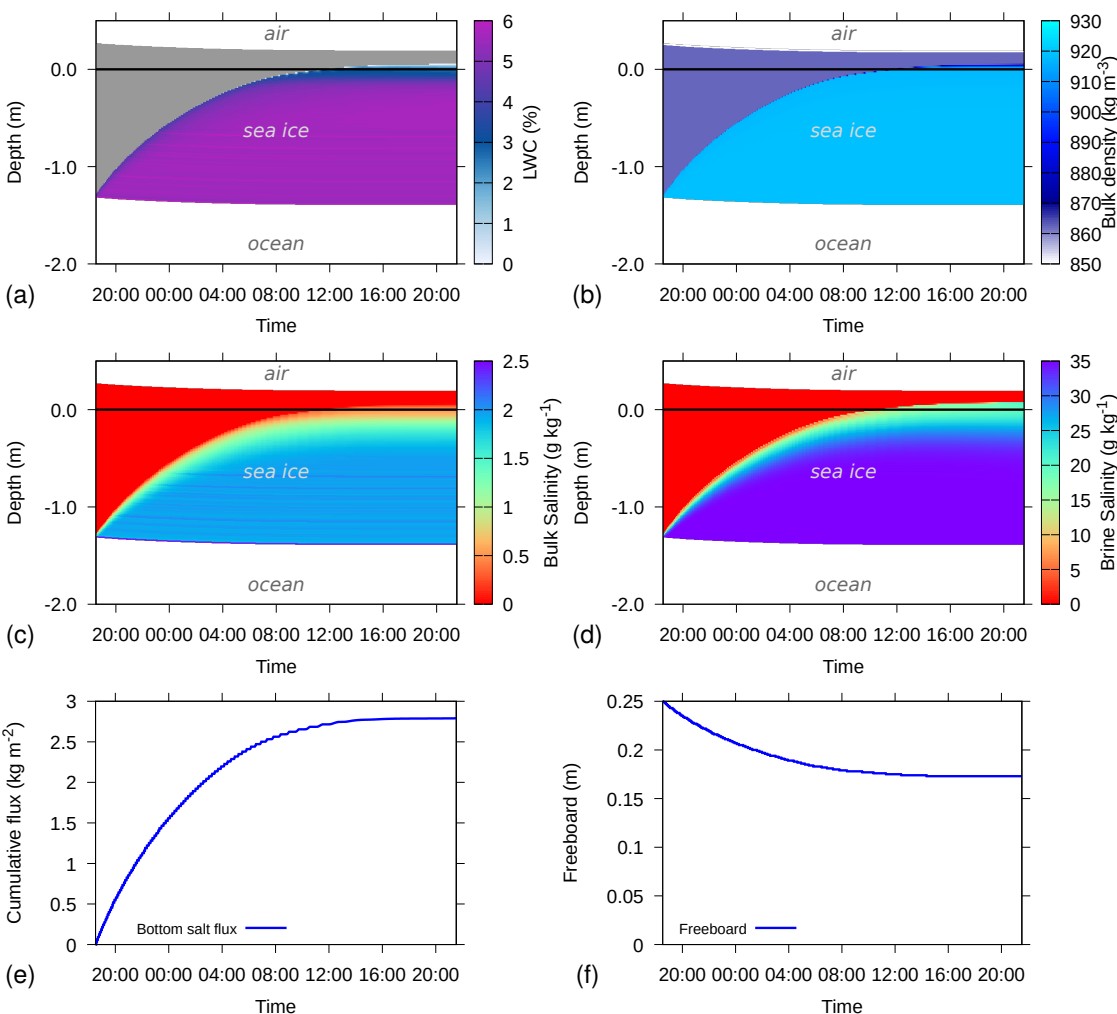

**Figure 4.** Example simulation where initially dry, porous freshwater ice with 94% volumetric ice content is placed into ocean water with a salinity of 35 g kg$^{-1}$. Shown are (a) liquid water content (LWC), (b) bulk density, (c) bulk salinity, (d) brine salinity, (e) cumulative salt flux at the bottom of the sea ice and (f) freeboard. In (a), dry parts of the ice are colored grey. In (a), (b), (c), and (d), the depth on the y-axis is relative to sea level, i.e., sea level is 0 and indicated by the solid black line.

upon installation of the Snow Buoy. The sea level as calculated from hydrostatic balance is indicated by the solid line. For the snow depth driven simulations, the sea level stays below the snow – ice interface, indicating that freeboard is positive during the whole simulation period. However, for the precipitation driven simulations, there is more snowfall, which causes negative freeboard from October onward.

5    Fig. 5c shows the measured sea ice temperatures from the corresponding IMB. Note that for this IMB, the thermistor chain does not extend above the initial 2 cm snow layer on top of the sea ice, such that the time evolution of the snow cover is not

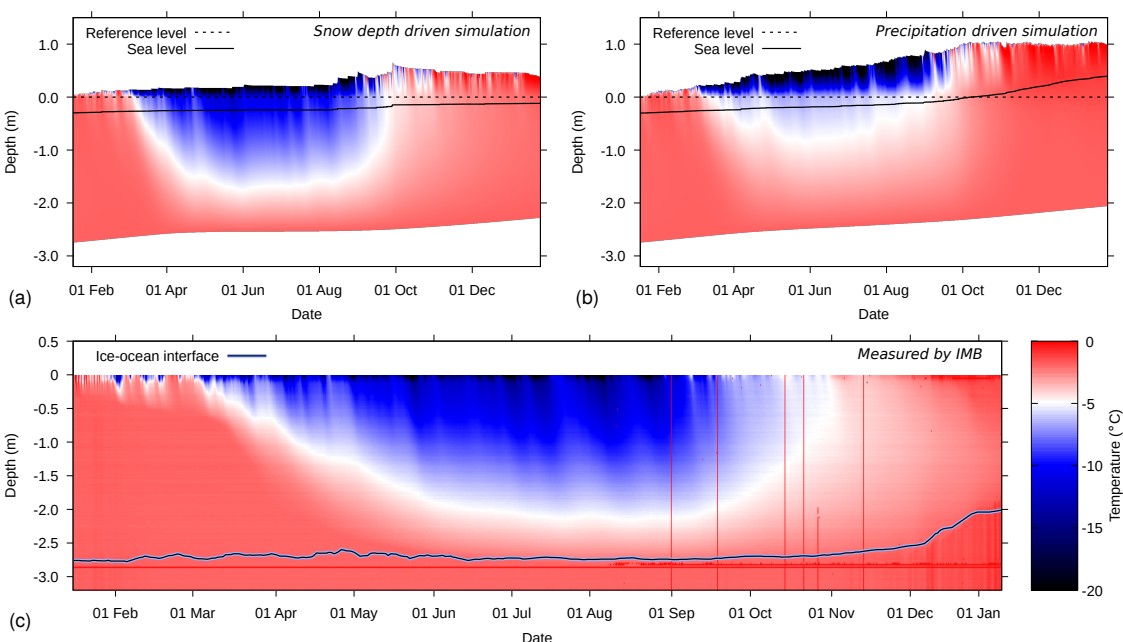

**Figure 5.** Snow and ice temperatures for Snow Buoy 2016S31 / IMB 2016T41, for (a) simulations driven by in-situ measured snow depth, (b) simulations driven by ERA5 precipitation, and (c) measured temperatures by the IMB. The depth on the y-axis is defined relative to the snow – ice interface, as determined upon installation (dashed line). In (a) and (b), the solid line denotes the sea level as determined by the simulations. In (c), the blue/black line denotes the ocean – ice interface, as determined from the IMB measurements.

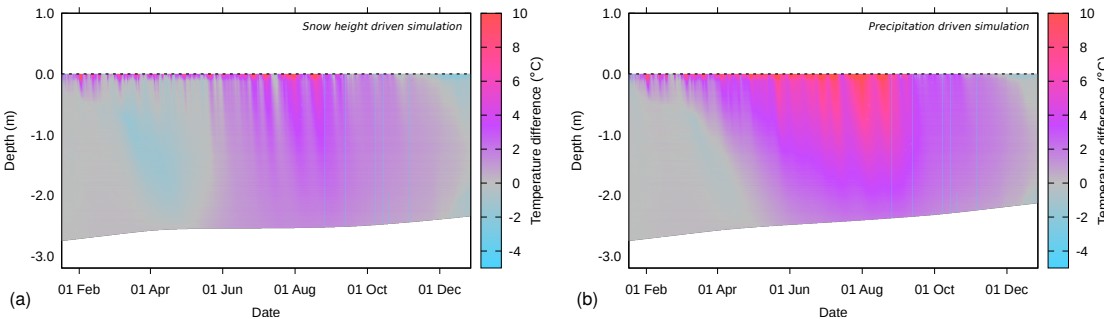

**Figure 6.** Difference between simulated and measured snow and ice temperatures for Snow Buoy 2016S31 / IMB 2016T41, for (a) simulations driven by in-situ measured snow depth, (b) simulations driven by ERA5 precipitation. The depth on the y-axis is defined relative to the snow – ice interface, as determined upon installation.

recorded by the IMB. We find that the IMB confirms the strong cooling of the sea ice during the austral winter months, as found in the simulations, as well as the near-surface warming to the melting point of fresh water shortly before the last transmission by the buoy.

Fig 6 shows difference plots of measured and simulated ice temperatures. Fig 6a compares the snow depth driven simulations (i.e., Fig 5a minus Fig. 5c) and Fig 6b compares the precipitation driven simulations (i.e., Fig 5b minus Fig. 5c). Positive values denote an overestimated temperature by the model and vice versa. Note that the bottom of the sea ice in the simulations is typically above the ice – ocean interface in the IMB data.

The comparison shows that in the period March to May, the snow depth driven simulation slightly underestimates the sea ice temperatures, which suggests an overestimation of the initial cooling of the sea ice towards austral winter. Both simulations underestimate the lowest temperatures reached in winter by up to 6-15 °C near the snow – ice interface, which is located at the top of the thermistor chain. Similarly, incidental cooling in near surface sea ice layers in February and March are also underestimated. This could on the one hand indicate an overestimation of incoming energy or an underestimation of outgoing

energy in the forcing data or by the model. On the other hand, new snow density in polar regions is higher than in alpine snow covers, due to the impact of drifting snow (Groot Zwaaftink et al., 2013; Steger et al., 2017). Even though the new snow density in SNOWPACK is parameterized using wind speed (in addition to air temperature, surface temperature and relative humidity), and also the snow settling formulation employed here leads to higher densities under the influence of wind (Lehning et al., 2002b), near-surface (new) snow density is likely still underestimated. An underestimated (new) snow density would result in

an underestimated thermal conductivity and overestimated thermal insulation in the model, as thermal conductivity typically increases with increasing snow density. Heat from the sea ice part would then not be able to be sufficiently transported through the snowpack and exchanged with the atmosphere.

     The simulations driven by ERA5 precipitation show about twice as much snow accumulating on the sea ice (Fig. 5b) compared to accumulations determined from the snow height measurements (Fig. 5a). Also the reanalysis provides precipitation

(and thus snow accumulating) in the austral winter time, which is not found in the snow depth time series. This is not necessarily a bias in ERA5, because snow erosion by wind can keep the snow depth constant over extended periods of time. Furthermore, if new snow density is underestimated, snow depth is likely overestimated, even though snow settling reduces the discrepancy. Due to these factors, the total snow depth may be overestimated by the ERA5 input at the end of the simulation. Furthermore, the thick snow cover in the precipitation driven simulation better insulates the underlying ice, resulting in a stronger overestimation

of ice temperatures compared to the snow height driven simulation.

     The thermistor chain is also used to determine the heat capacity by heating the chain for 1 or 2 minutes and analyzing the temperature response. By combining the absolute temperatures and the heating rates, the ice – ocean interface has been manually determined and is shown in Fig. 5c. The IMB data confirm the modelling result that the strong negative energy balance at the top of the snow – sea ice system during austral winter has not resulted in an ice thickness increase. The warmer

sea ice in the precipitation driven simulations resulted in a thinning of the ice by the ocean heat flux, which is not confirmed by the IMB data. The decrease in ice thickness in December 2016 is not reproduced by either one of the simulation setups. The trajectory of the buoy shows a marked change in drift direction (Fig. 1), changing from a northward to a southward drifting course during this period. This change of direction may have been accompanied by an intrusion of warm ocean water below the sea ice and an increased ocean heat flux.

LWC, bulk and brine salinity, as well as the bottom salt flux for the simulations driven by measured snow depth are shown in Fig. 7. LWC (Fig. 7a) shows a strong reduction in austral winter due to the freezing brine, causing brine salinity to increase (Fig. 7c). The snow is dry most of the time, except towards the end of the simulation when meltwater percolates downward. Furthermore, we find a thin layer with low values of LWC just above sea level. This is caused by capillary forces causing

5   upward motion of sea water above sea level.

Fig. 7b shows that the bulk salinity of the sea ice hardly changes over the course of the simulation, whereas the brine salinity (Fig. 7c) clearly shows a relationship with the temperature. This reflects the prescribed thermal equilibrium between brine and the ice, assuming that the brine is at melting temperature. Fig. 7d shows that the added weight of the accumulating snow pushes the sea ice deeper into the ocean, increasing the pressure head at the bottom of the sea ice. This leads to an influx of saline

10   water, even though the sea level remains below the snow – ice interface. Combined with rising temperatures, a thin layer with increased bulk salinity and increased LWC forms around and just below sea level around the 1st of October.

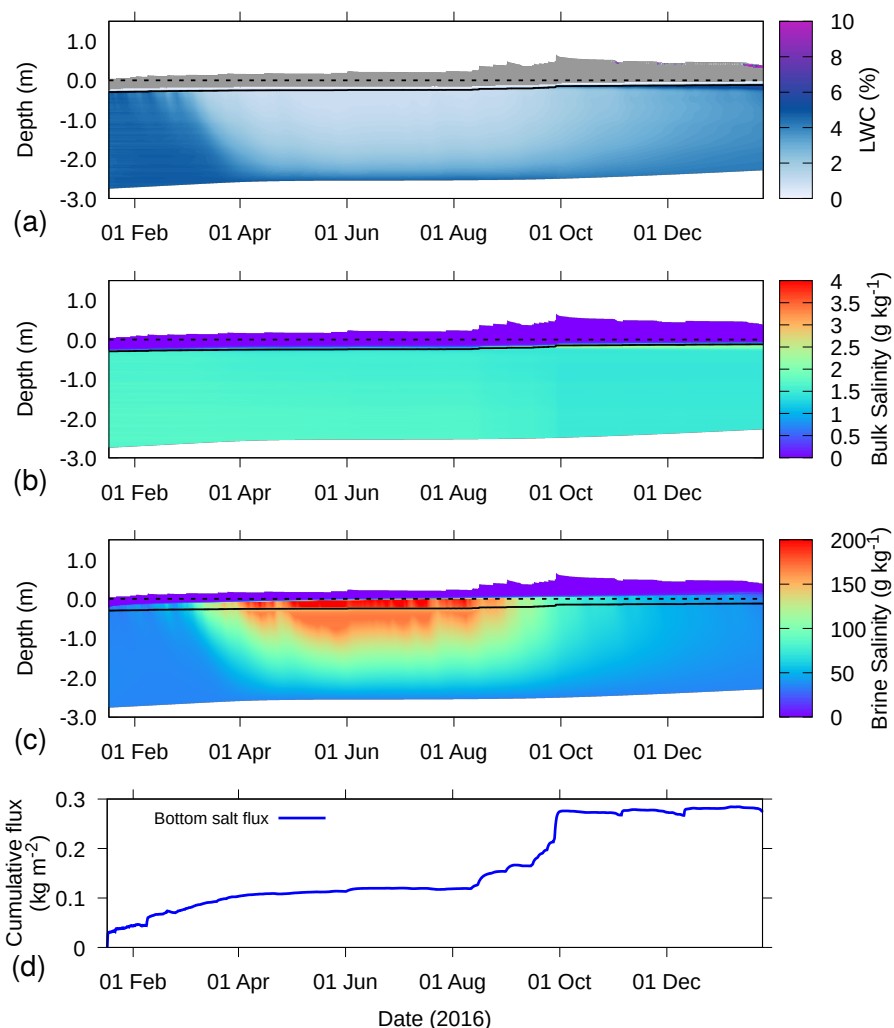

**Figure 7.** Example simulation for Snow Buoy 2016S31, where measured snow depth was used to derive the precipitation events, for (a) LWC, (b) bulk salinity, (c) brine salinity and (d) cumulative salt flux at the bottom of the sea ice. In (a), dry parts of the snow – sea ice system are colored grey. In (a), (b), and (c), the depth is defined relative to the snow – ice interface, as determined upon installation (dashed line). The solid line denotes the sea level as determined by the simulation. In (d), an increasing cumulative flux denotes inflow and vice versa.

### 4.3 Flooding and Superimposed Ice Formation

Figs. 8 and 9 show an example simulation for Snow Buoy 2014S12. The simulations were driven by the snow depth measurements from the buoy. Upon installation of the buoy, the snow depth was referenced to the sea ice surface. This reference level is shown by the dashed line. Due to basal ice melt and growth, as well as additional snowfall, the simulated sea level became higher w.r.t. to the snow depth sensor, as indicated by the solid line. This is congruent with a negative freeboard. It indicates that significant flooding occurred, as also evidenced by the LWC in Fig. 8b and associated high bulk salinity (Fig. 8c). Fig. 9a shows that the bulk density of the flooded part is similar to the underlying ice density, illustrating the depletion of pore space. Fig. 9b shows an increase in ice content in the flooded layers, indicating that a substantial amount of the sea water flooding the snow refroze, adding considerable ice mass.

The simulated temperature (Fig. 8a) shows the two austral winters with low temperatures, and the two austral summers with temperatures close to 0 °C. Interestingly, the low temperatures in the first austral winter impacted the part below sea level stronger than the second austral winter, as demonstrated by the lower temperatures in that part of the domain. During the second austral winter, flooding in the simulation increased the liquid water content of the snow and consequently, much of the energy loss by the sea ice in this period was balanced by the energy release from refreezing liquid water, rather than decreasing temperatures.

Fig. 8e shows that flooding also leads to a strong influx of salt to the snow-sea ice system. The flooding saturates the snow, which has significant pore space compared to the sea ice. Therefore, snowfall events of similar magnitude have different effects on the salt influx, depending on whether or not flooding occurs at the time.

Note that the flooding, as depicted in simulations with the Richards equation coupled to the transport equation, is governed by the hydraulic conductivity of ice. In cold ice, hydraulic conductivity can be so low that negative freeboard remains for extended period of times, even without flooding. On the other hand, flooding may also be triggered by deformation and cracking of the sea ice, combined with lateral flow effects. However, Maksym and Jeffries (2000) have shown that the simpler assumption (i.e., negative freeboard will trigger flooding) can already yield satisfying results. In the model, the maximum ice content is fixed to $0.99$ m$^3$ m$^{-3}$ and is typically lower, such that hydraulic conductivity is generally large enough for instantaneous flooding.

The brine salinity in Fig. 8d shows spurious oscillations. These originate from the lower boundary and could be caused by the strong transition of brine salinity to ocean salinity. Also, the oscillations are partly attributed to the maximum allowed ice content of $99\%$ in the simulations, as they occur in the same area. The exact numerical mechanism and a possible solution is currently unknown.

Figs. 9c and 9d show the snow microstructure as simulated by the model. Even though validation for this specific floe is not possible, we find several features consistent with other field observations. For example, a wet slush layer (coloured red) at the interface between snow and ice is visible, triggered by capillary suction and flooding. This is also reported in Nicolaus et al. (2009) and Arndt and Paul (2018) for the same geographical region. Even though those field observations report the presence of depth hoar layers in snowpacks, particularly in deeper snowpacks (exceeding 30 cm), the simulations seem to show deeper depth hoar layers (coloured blue) than reported in field observations. The field observations often report wind

slabs or other hard layers in between depth hoar layers. Simulating those kind of layers is a well-known problem for snowpack models (Domine et al., 2019).

The grain size shown in Fig. 9d ranges from 2-2.5 mm in the depth hoar layers at the base of the snowpack to 0.5-1 mm near the top of the snowpack. These simulated grain sizes corresponds to the range of values reported by the field studies listed earlier (1-5 mm), yet exact validation remains difficult.

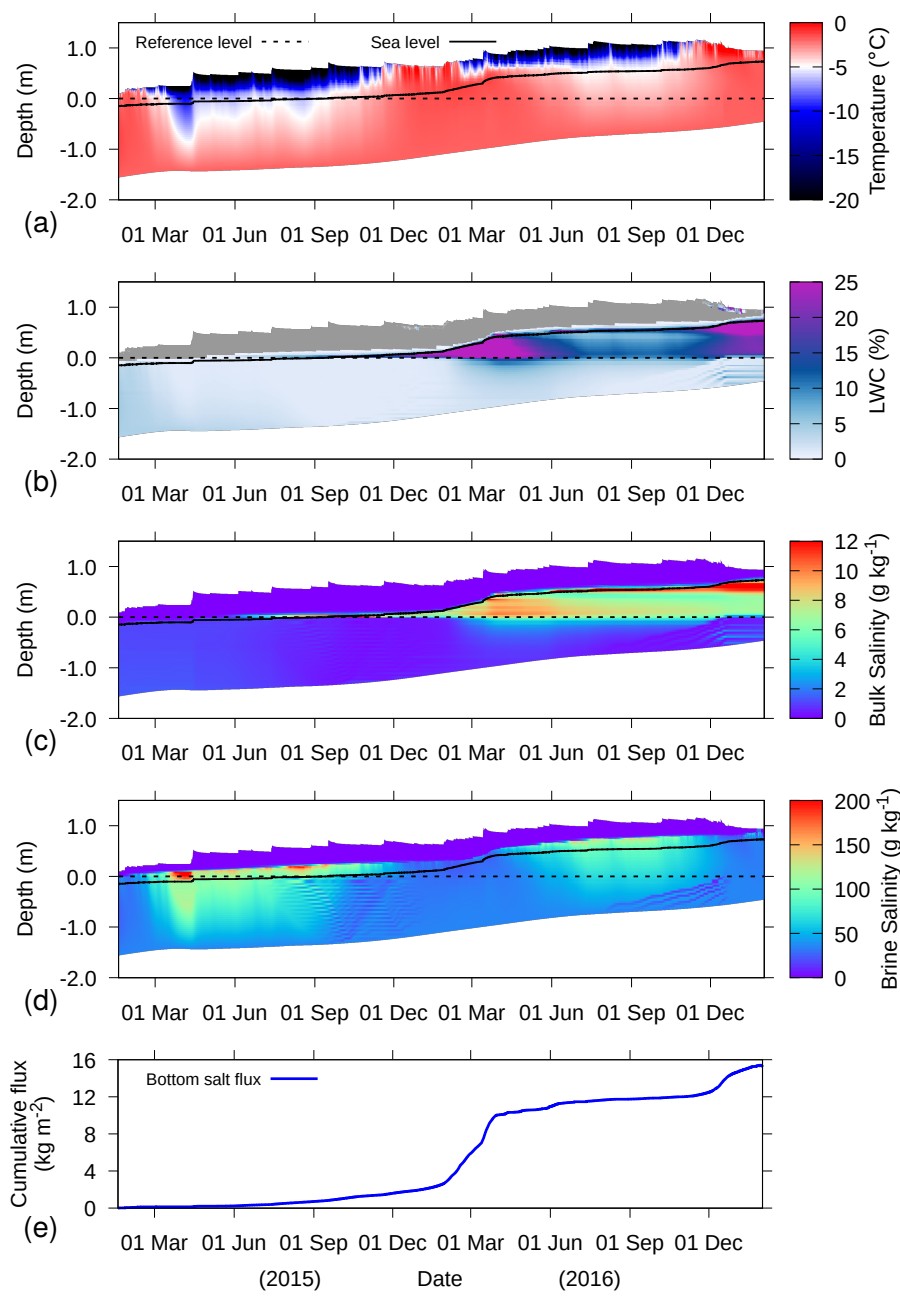

**Figure 8.** Example simulation for buoy 2014S12, where measured snow depth was used to derive the precipitation events, for (a) temperature, (b) LWC, (c) bulk salinity, (d) brine salinity and (e) cumulative salt flux at the bottom of the sea ice. In (b), dry parts of the snow – sea ice system are colored grey. In (a), (b), (c), and (d), the depth is defined relative to the snow – ice interface, as determined upon installation (dashed line). The solid line denotes the sea level as determined by the simulation. In (e), an increasing cumulative flux denotes inflow and vice versa.

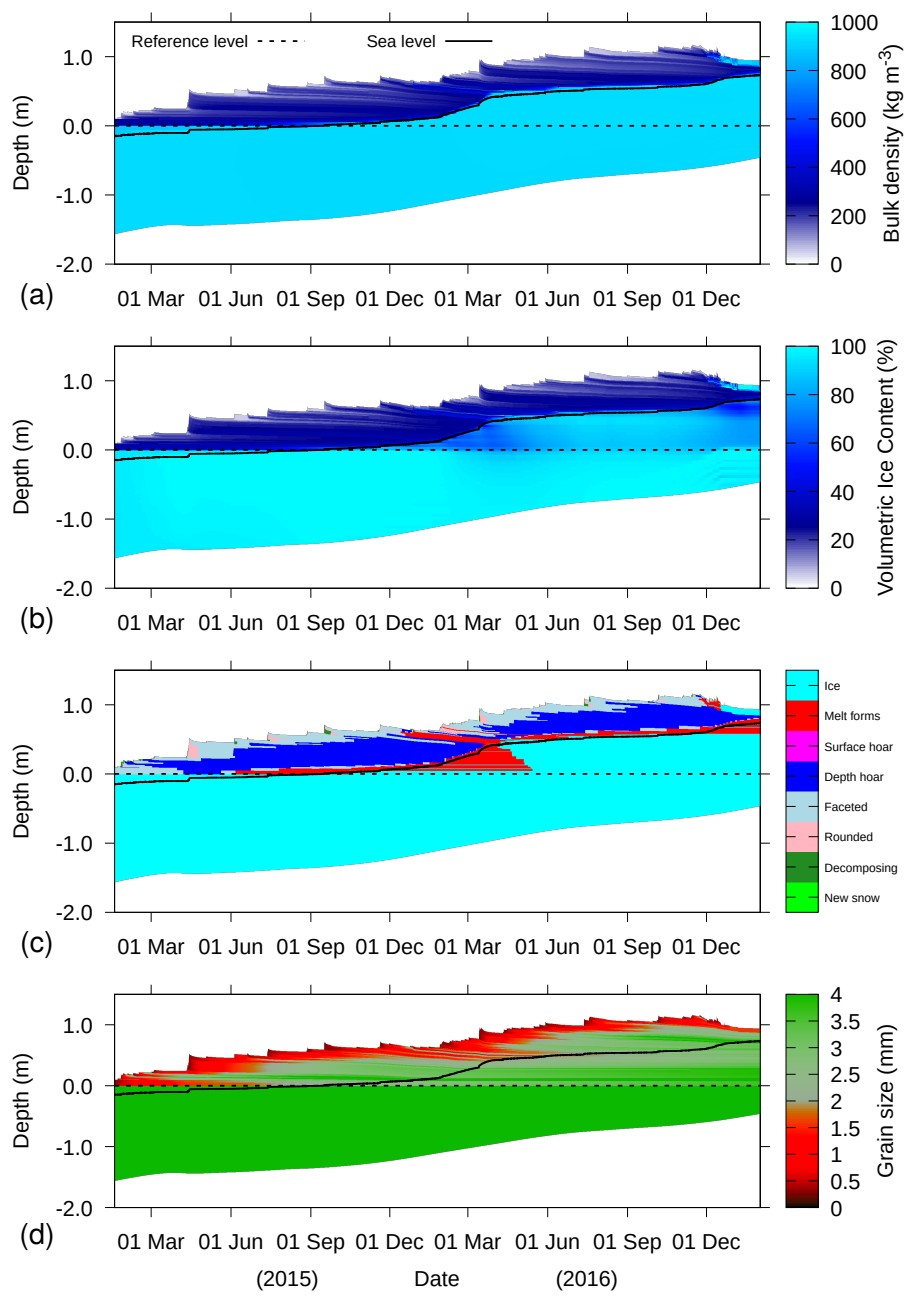

**Figure 9.** Example simulation for buoy 2014S12, where measured snow depth was used to derive the precipitation events, for (a) total bulk density, (b) volumetric ice content, (c) grain type and (d) grain size. The depth on the y-axis is defined relative to the snow – ice interface, as determined upon installation (dashed line). The solid line denotes the sea level as determined by the simulation.

## 4.4 Forced Warming and Cooling

Warming conditions increase the brine volume and the hydraulic conductivity of the sea ice, but also can cause freshwater percolation from snow melt. Similarly, cooling conditions decrease brine volume and increase brine salinity and density. When brine channels allow, the dense brine may drain from the sea ice.

To test how the model reacts to continuous warming or cooling conditions, we used Snow Buoy 2016S31 for two experiments where we forced constant warming and constant cooling conditions by modifying the meteorological driving data starting April 1, 2016. To force warming conditions, we prescribed a constant air temperature of +5 °C, a relative humidity of $80\%$, a wind speed of 3 m s$^{-1}$ and an incoming shortwave radiation of 300 W m$^{-2}$.

Fig. 10 and 11 show the simulation result for the warming experiment. Fig. 10 shows that as soon as melting conditions were enforced (starting April 1st), the snowpack very quickly reached melting temperature. The fresh water percolating as a result from snow melt first started accumulating on below-freezing sea ice with low porosity (see Fig. 10b). This process is visible in April and leads to a thin layer of superimposed ice (Fig. 11b and 11c). Starting April 15, the bulk density of the snow – sea ice system was homogeneously around ice density, showing that the pore space has been depleted (11a).

The fresh water started flushing the ice around April 10, leading to a rapid reduction of bulk and brine salinity (Fig. 10c,d), due to the outflow of saline water at the bottom of the sea ice (Fig. 10e). When the brine salinity decreases, water freezes and the permeability of the sea ice is low compared to the surface melt. We find that meltwater accumulates on the top of the sea ice (Fig. 10b), which can be interpreted as the formation of a melt pond.

The sea ice thins continuously upon continued warming (Fig. 11b), until the ice has melted and the melt pond disappears. At this point, the simulations showed strong instabilities in the bottom salt flux, such that the cumulative flux in Fig. 10e is only shown for the time period with ice below the liquid water. Note that the melt pond in the model consists of a little bit of ice (Fig. 11b), which results from the SNOWPACK numerics.

Fig. 12 shows an example where cooling conditions were enforced. Similar to the warming example, the meteorological forcing conditions were replaced from April 1 onward by setting a constant air temperature of -30 °C, a wind speed of 1 m s$^{-1}$ and no incoming shortwave radiation. The ocean heat flux was set at 8 W m$^{-2}$. As soon as cooling conditions were present, a freezing front progressed through the sea ice. The interface between the sea ice and the ocean remained at the freezing point of ocean water, while the sea ice froze and increased thickness.

With the decrease in temperature, brine salinity increased (Fig. 12d). This is achieved by freezing the liquid water, as shown by the decrease in LWC (Fig. 12b). An increase in brine salinity also increases the density of the brine. This may lead to flushing of the sea ice, when the heavy brine moves downward and is replaced by lighter ocean water. However, in our simulations, there is only a very small outflow of salinity at the bottom (Fig. 12e) and the bulk salinity remains approximately constant (Fig. 12c). This result shows that without a description of the convective processes in sea ice resulting from cooling (Griewank and Notz, 2013), the salinity depletion found due to cooling is strongly underestimated by this model approach.

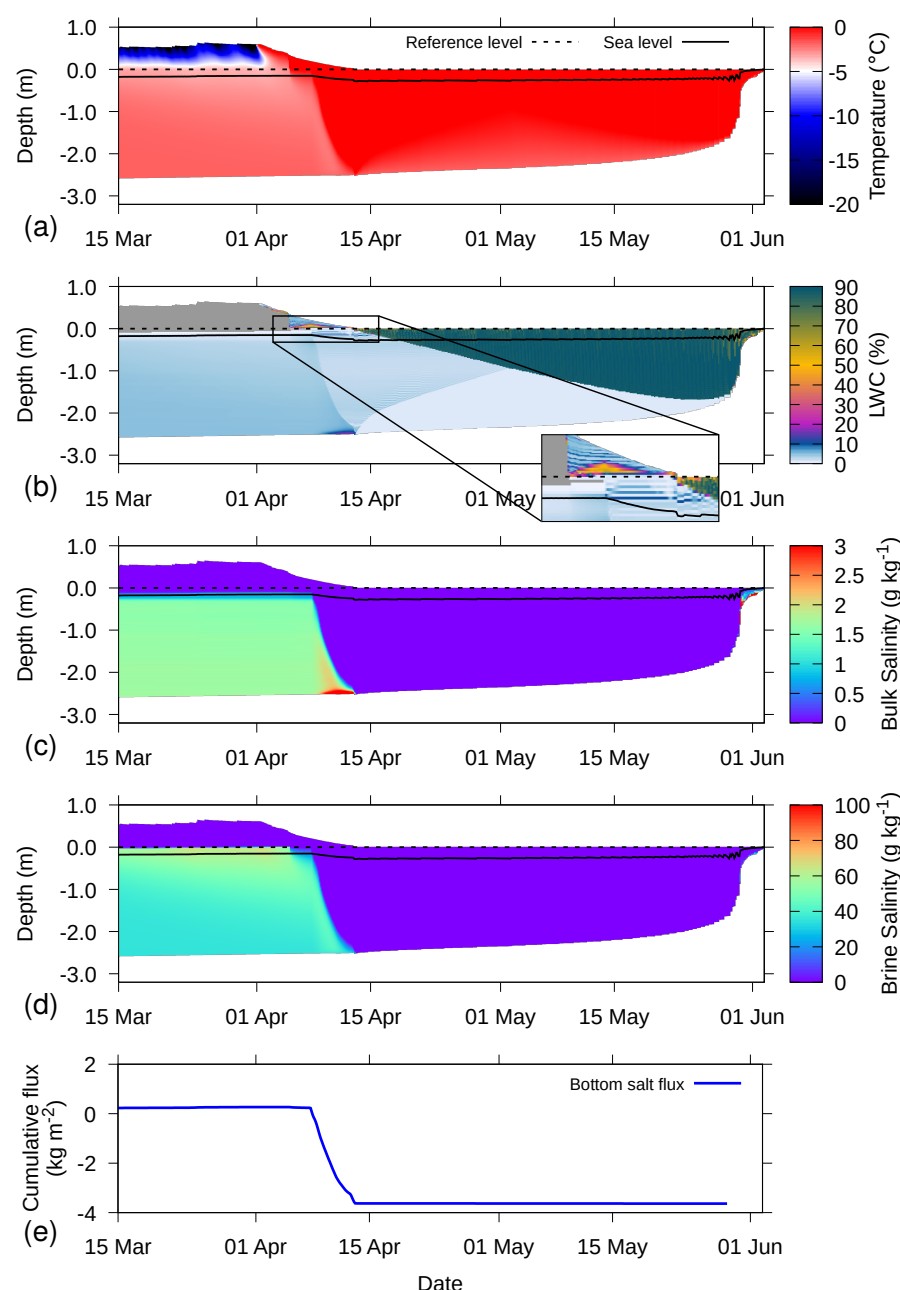

**Figure 10.** Simulation for 2016S31, where from April 1 onward, melting conditions were enforced, for (a) temperature, (b) LWC, (c) bulk salinity, (d) brine salinity and (e) cumulative salt flux at the bottom of the sea ice. In (b), dry parts of the snow – sea ice system are colored grey and an inset shows the water percolating through the snow and ponding on the snow – ice interface. In (a), (b), (c), and (d), the depth on the y-axis is defined relative to the snow – ice interface, as determined upon installation (dashed line). The solid line denotes the sea level as determined by the simulations. An increasing cumulative flux denotes inflow and vice versa.

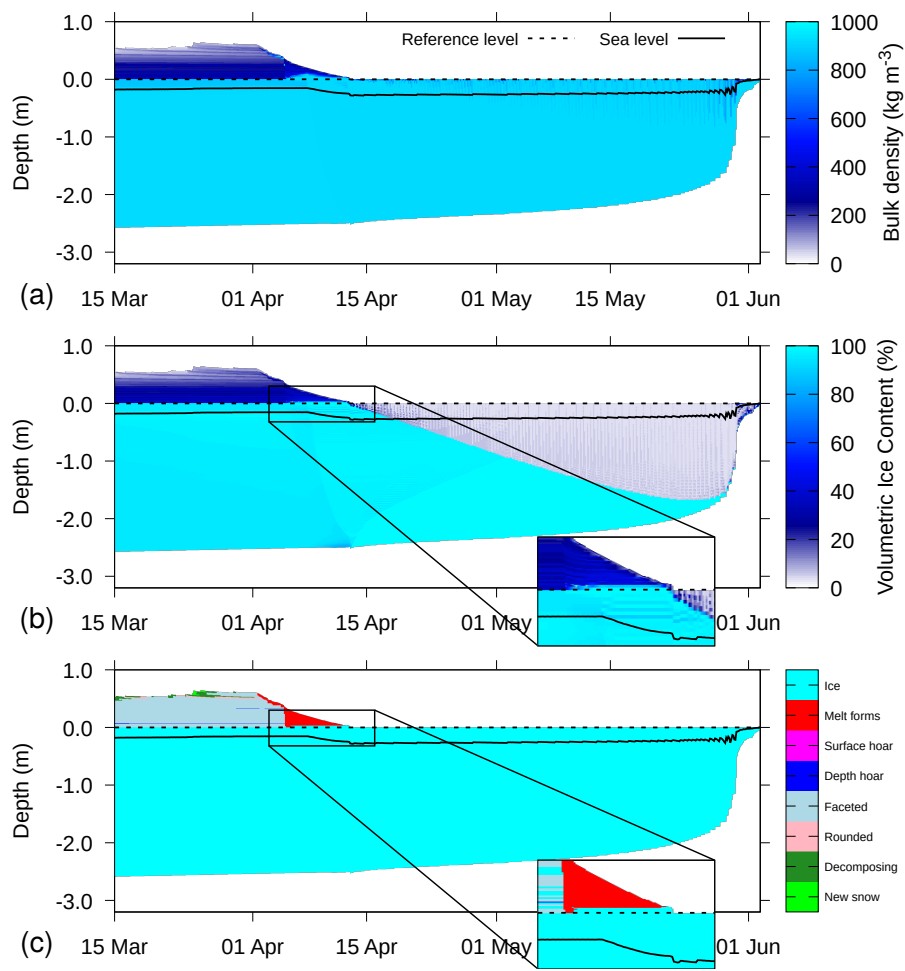

**Figure 11.** Simulation for 2016S31, where from April 1 onward, melting conditions were enforced, for (a) total bulk density, (b) volumetric ice content, and (c) grain type. In (b) and (c), an inset shows the superimposed ice formation on the snow – ice interface. The depth on the y-axis is defined relative to the snow – ice interface, as determined upon installation (dashed line). The solid line denotes the sea level as determined by the simulations.

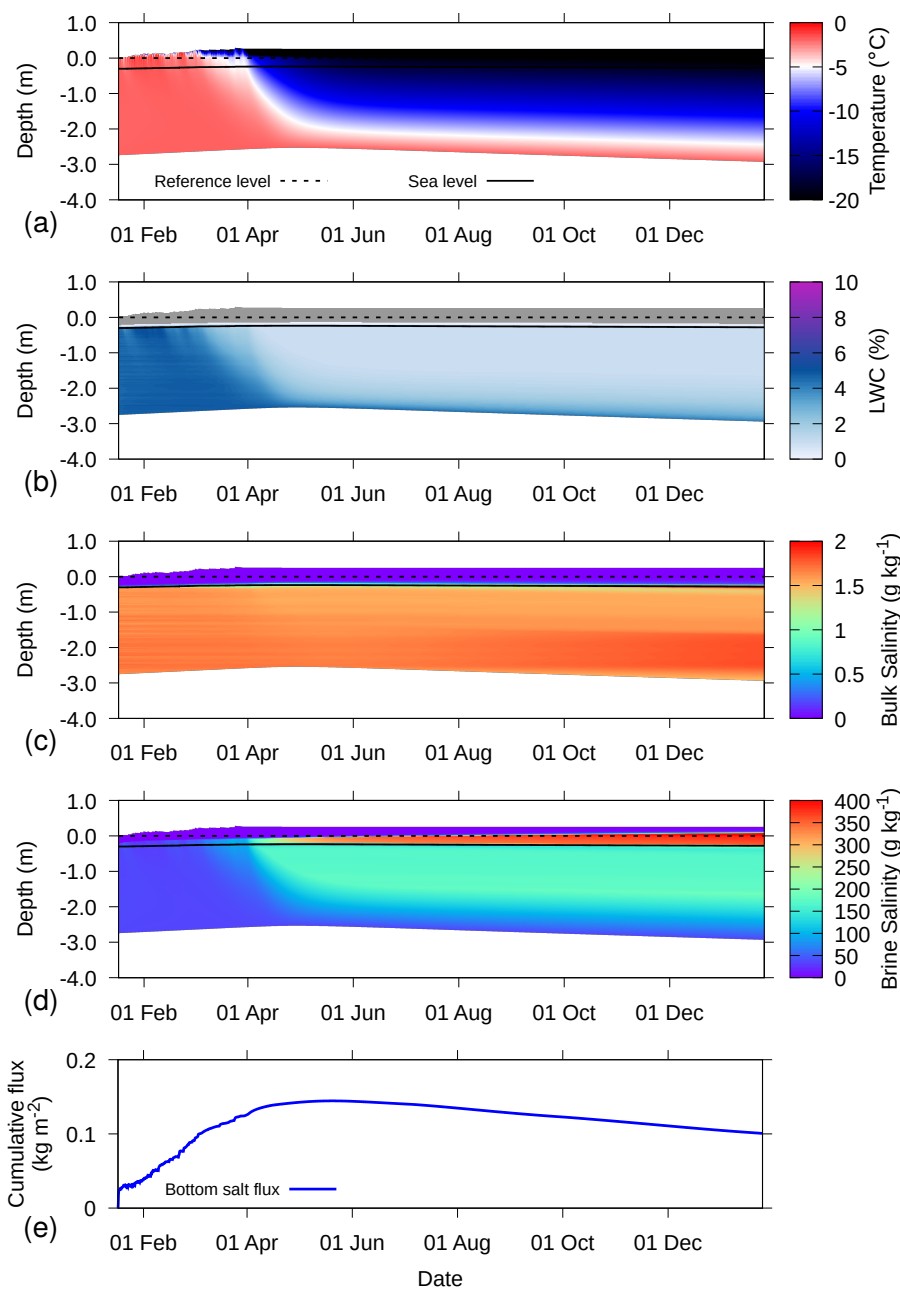

**Figure 12.** Simulation for 2016S31, where from April 1 onward, cooling conditions are enforced, for (a) temperature, (b) LWC, (c) bulk salinity, (d) brine salinity and (e) cumulative salt flux at the bottom of the sea ice. In (b), dry parts of the snow – sea ice system are colored grey. In (a), (b), (c), and (d), the depth on the y-axis is defined relative to the snow – ice interface, as determined upon installation (dashed line). The solid line denotes the sea level as determined by the simulations. An increasing cumulative flux denotes inflow and vice versa.

## 4.5 Thin Ice

A final test is run by starting with only 2 cm of ice and constant atmospheric conditions to simulate thin ice evolution. The atmospheric conditions were set as -10 °C air temperature, 100% relative humidity, no wind speed, no incoming solar radiation and a constant incoming longwave radiation of 230 W m$^{-2}$. The ocean heat flux was set to 0 W m$^{-2}$.

The simulations were run for 1 month, which resulted in approximately 50 cm ice growth (Fig. 13). The temperature distribution (Fig. 13a) shows a very strong gradient, as the bottom temperature is forced to the ocean water temperature, whereas the surface cools from radiation loss and sensible and latent heat exchange. The relatively warm sea ice compared to the air temperature results in a latent heat flux directed to the atmosphere, even though relative humidity is 100%. The evaporation at the top of the sea ice leads to an outflow of fresh water at the top of the snowpack, resulting in an accumulation of salt near the surface (e.g., Kaleschke et al., 2004; Domine et al., 2005). This is found in Fig. 13b and Fig. 13c in a salty slush layer at the surface with high LWC and high bulk salinity. The brine salinity (Fig. 13d) mimics the temperature distribution (Fig. 13a), because of the forced thermal equilibirum with the brine by the model. When the ice is very thin, the evaporation at the top of the ice causes an influx of salt at the bottom (Fig. 13e). The transport of salt from below decreases with increasing ice thickness, as capillary forces are not strong enough anymore to bridge the freeboard.

## 5 Outlook

Here, we showed crucial modifications to the SNOWPACK model with the primary goal of simulating the snow covering sea ice. As we initially focus on the Southern Ocean, the modifications centered around liquid water percolation in the snow, and flooding with ocean water of the snow layer. These are crucial processes to simulate for interpreting snow height measurements.

Nevertheless, these first sets of simulations revealed several directions for future improvements. First of all, the used relationship between temperature and brine salinity (Eq. 2) is only valid for temperatures close to the melting temperature of water. Other relationships have been proposed (Vancoppenolle et al., 2019) and may be important to include for more accurate simulations.

Brine dynamics in sea ice is a very complex process. Particularly gravity drainage of brine is complex to simulate (Notz and Worster, 2009) and can have a profound influence on bulk salinity profiles. Typically, a decreasing bulk salinity is found with increasing floe thickness, due to gravitiy drainage (Kovacs, 1996). The current model framework is not able to reproduce this. Furthermore, cooling of thin ice during the freezing process increases the pressure in the brine pockets, which can cause upward brine migration and higher salinity near the sea ice surface. Our simulations show this as a result of an evaporative flux. However, the increased pressure from freezing may potentially be described by an additional term for the capillary pressure (Eq. 4).

The Crank-Nicolson scheme used for the transport equation causes some numerical instabilities, particularly when the volumetric ice content is at the prescribed upper limit of $99\%$. This is a known problem with the Crank-Nicolson scheme (Østerby, 2003), and could be mitigated by using other numerical schemes.

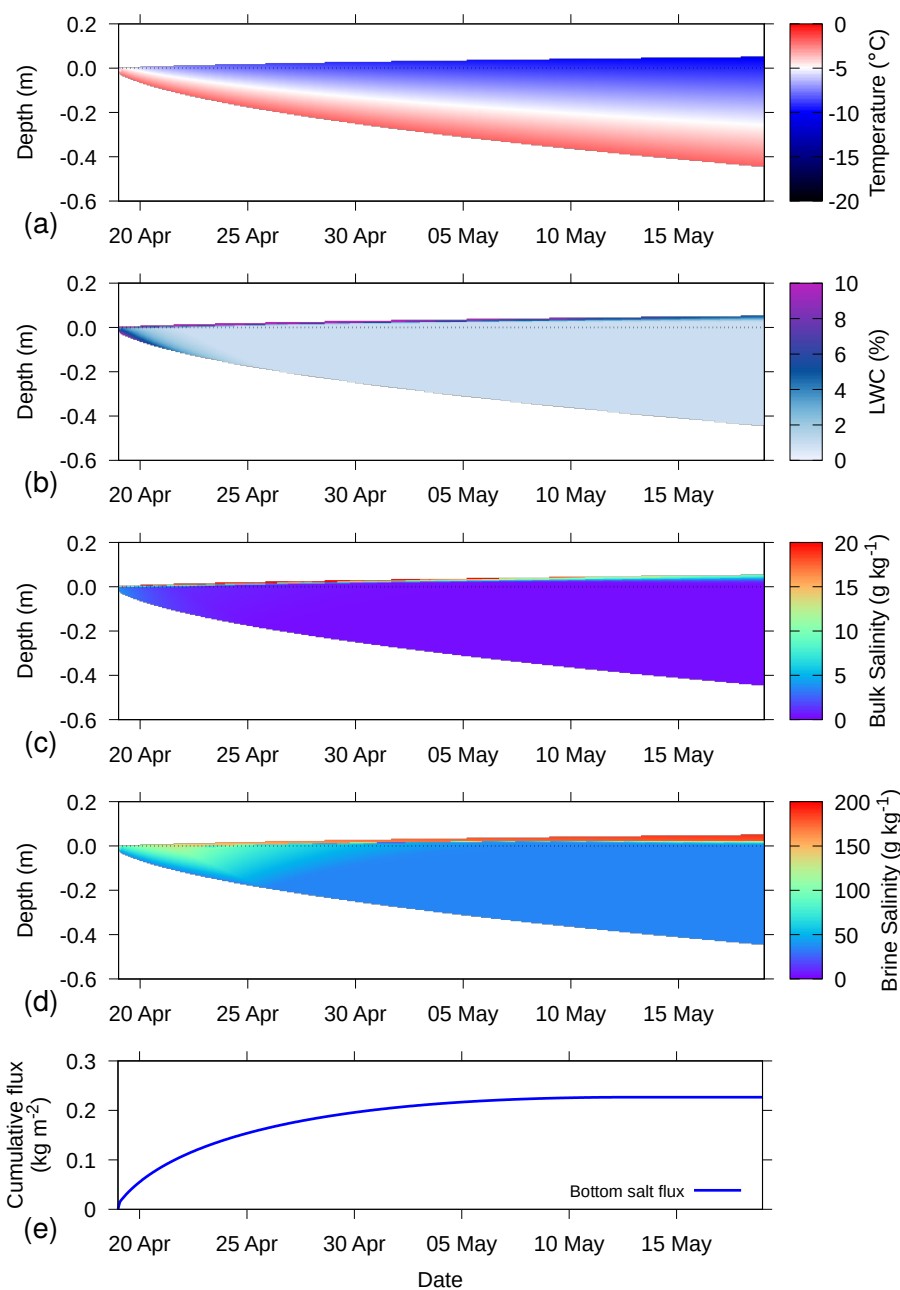

**Figure 13.** Simulation for ice growth of thin ice, for (a) temperature, (b) LWC, (c) bulk salinity, (d) brine salinity and (e) cumulative salt flux at the bottom of the sea ice. In (a), (b), (c), and (d), the depth on the y-axis is defined relative to sea level. An increasing cumulative flux denotes inflow and vice versa.

## 6 Conclusions

We introduced a series of modifications to the physics-based, multi-layer SNOWPACK model for simulations of the snow – sea ice system. The thermodynamic description in the model was modified to account for the varying melting point of ice based on salinity and adding domain restructuring to allow basal ice growth. Water transport through the snow – sea ice
system is described by the Richards equation, which describes water flow in porous media for the full range from saturated conditions (Darcy law) to unsaturated conditions. This equation is coupled to a concentration equation for salinity. With the adapted model, we explicitly describe several aspects of brine dynamics, such as flooding, superimposed ice formation and the percolation of fresh water from snow melt, flushing the sea ice. Using examples of snow and ice buoys from the Southern Ocean, we show that the model reproduces those processes with plausible detail. The model formulations allow for a certain
amount of drainage of dense brine, but the process is largely underestimated compared to what is known from literature, as convective brine transport is, thus far, not described by the model.

The snow microstructure descriptions previously developed in the SNOWPACK model can now be applied for sea ice conditions as well. The model is able to simulate the temporal evolution of snow density, grain size and shape and snow wetness over the life span of an ice floe. We find abundant depth hoar layers and melt layers, as well as superimposed ice
formation due to flooding and percolation. The detailed snow microstructure evolution has the potential to be used to improve remote sensing retrieval algorithms to assess snow depth and ice thickness from space and driving radiative transfer models such as the Snow Microwave Radiative Transfer model SMRT (Picard et al., 2018).

Driving the simulations using reanalysis model output seems to work well, apart from uncertainties in estimating the ocean heat flux from below and estimating precipitation amounts. The ability of SNOWPACK to use the in-situ snow depth to de-
termine snow fall amounts was found to be useful for assessing the mass balance, but is difficult to upscale due to limited measurement data from polar regions. The simulations based on Snow and IMB Buoy data demonstrate, however, the importance of such remote data collection systems for modelling.

*Code and data availability.* The SNOWPACK model and the MeteoIO meteorological preprocessing library (Bavay and Egger, 2014) needed to run SNOWPACK are available under a LGPLv3 license under https://models.slf.ch. The source code of the version used for the sim-
ulations presented in this study is available in the Online Supplement, and corresponds to revision 2508 of MeteoIO (https://models.slf.ch/svn/meteoio/trunk) and revision 1799 of SNOWPACK (https://models.slf.ch/svn/snowpack/branches/dev). The source code, input and configuration files, as well as run, postprocessing and plotting scripts for the example simulations in this study are also available in the Online Supplement. The website https://niviz.org/ can be used to visualize the SNOWPACK output files.

The data for Snow Buoys 2014S12 and 2016S31 can be acquired via doi: 10.1594/PANGAEA.875272 (Nicolaus and Schwegmann, 2017) and
doi: 10.1594/PANGAEA.875287 (Arndt et al., 2017), respectively. IMB data from Buoy 2016T41 are available at http://www.meereisportal. de, direct link: http://data.meereisportal.de/gallery/index_new.php?active-tab1=method&buoytype=TB®ion=s&buoystate=inactive&submit3= display&lang=en_US&active-tab2=buoy. ERA5 data can be accessed via doi: 10.5065/D6X34W69 (ERA, 2017).

*Author contributions.* NW developed the model code and performed the simulations, LR performed code testing, LR, NM, KL, LK, MN and ML contributed to model conceptualization and design. NW prepared the manuscript with contributions from all co-authors.

**Acknowledgements**

We thank the captain, officers and crew of R/V Polarstern for their support during the campaigns to deploy the Snow Bouys
5   and IMBs. Louisa von Hülsen from the Alfred Wegener Institute is acknowledged for providing preliminary IMB analysis data. N.W. was supported by the Swiss National Science Foundation (SNSF), grant no. P2ELP2_172299. This work was additionally supported by the US National Science Foundation (NSF) grant no. OPP-1142075 and Swiss National Science Foundation grant no. PZ00P2_142684. We also thank the German Research Council (DFG) for funding the Snow cover impacts on Antarctic Sea Ice (SCASI) project within the framework of the priority program "Antarctic Research with comparative investigations in
10  Arctic ice areas" (grant no: NI 1096/5-1 and KA 2694/7-1). The Helmholtz infrastructure programs FRAM and ACROSS is acknowledged for funding the Snow and Ice Mass-balance Buoys. ERA5 data constitute modified Copernicus Climate Change Service Information [2018].

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
