# Peer review of "Version 1 of a sea ice module for the physics based, detailed, multi-layer SNOWPACK model"

_Geoscientific Model Development, 2019_

## Referee Comment (RC1) · Anonymous Referee #1 · 11 Jul 2019

In the manuscript: "Version 1 of a sea ice module for the physics based, detailed, multi-layer SNOWPACK model", an extension of the physical based SNOWPACK model to sea ice is presented. Parts of the model equations are adapted to account for salinity. In addition, other relevant processes like sea ice growth and melt, flooding and brine dynamics are included. The authors discuss the performance of the model comparing it to snow buoy and IMB measurements over Antarctic sea ice. Overall, this manuscript is well written and has a clear structure. Modeling the evolution of snow on sea ice is a challenging task and the here presented model extensions provide important progress for the snow on sea ice modeling community. I believe that this model will be widely used in future research, especially if it will be coupled to SMRT like the authors state in their conclusions. However, the presentation and the discussion of the results are partly

too short and important details are missing. Therefore I recommend the manuscript to be published after major revisions. I have three general comments that I would like the authors to address.

General comments (GC):

GC1: In the description of the experiments, important details about the model initialization are missing. In SNOWPACK, the snow is described by grain parameters like dendricity, sphericity and bond radius. These parameters are hard to obtain from snow pit or snow buoy measurements and I wonder what the authors assumed in the model initialization (so far, it is not mentioned in the manuscript). A discussion about these parameters and their influence on the results would be very helpful for further SNOWPACK users. In addition, the authors should provide the full initialization profiles and SNOWPACK settings (I guess this can be provided in the appendix).

GC2: In general, the motivation and discussion of the different experiments (section 4.1 to 4.5) are too short and need to be extended. Some features, which are visible in the figures are not or only poorly discussed. I encourage the authors to carefully read my specific comments for the single chapters.

GC3: It seems like the authors used the same color-scale range for all figures. In some figures, it is impossible to distinguish between different values since the color-scale goes far beyond the actual values in the figure. I recommend to use individual, representative color-scales for each figure.

Specific comments:

Section 3.1: Please improve figure 1.

Section 4.1: From figure 2, it looks like it takes less than two hours until the 1.58 m thick sea ice is saturated. Is this realistic? It seems very fast to me. The caption for figure 2 (a) is confusing. I guess you mean "dry sea ice" instead of "dry snow"?. In figure 2 (a) to 2 (d) the color-scale range need to be improved.

[Figure]

Section 4.2: Please increase the y-axis in figure 3 (b). Please mention what the blue line in figure 3 (c) is (I guess the ice/ocean interface?). In figure 3 (d), a difference of 6-10°C is found close to the snow/ice interface. Do you expect the differences to be even higher at the top of the snow? As I understood, you use ERA5 incoming longwave radiation as a forcing data. So in principle, the differences shown in figure 3 (d) could also be caused by errors in the forcing data. It might be helpful to add a timeseries of the ERA forcing data to figure 3. I'm surprised that even at the ice/ocean interface, the difference can be up to 2°C. To my understanding, the temperature should always be around the freezing temperature, or not? The caption for figure 4 is wrong (there is no temperature plot). Please adapt the color-scale for figure 4 (b).

Section 4.3: It is not clear to me, what the authors want to demonstrate in this section. I therefore encourage the authors to write a short motivation at the start of the experiments (This also applies for the subsequent sections). In Figure 5 (a) to 5 (c), vertical stripes are visible at the end of the simulation, which look like numerical instabilities. Are these related to the bucked scheme or to a too big timestep?

Section 4.4 In figure 6 (b) and (d), horizontal stripes are visible in the ice in the middle and at the end of the simulation, which look like numerical instabilities. To my understanding, these patterns should not appear since you are using the Crank-Nicolson scheme and the CFL criteria. Do you have an explanation for this phenomena? Please also discuss if these instabilities could have influenced the results of the simulations. In figure 6 (d), high brine salinity is simulated at the top of the snowpack. This must be wrong as I can't find any explanation how this could have happened? (especially since the snow was dry according to figure 6 (b)). Figure 7 (a) , (c) and (d) are not discussed in the text.

Section 4.5. Please extend the discussion of the experiments. The color-scale for figure 8 (a) needs to be adjusted.

Section 4.6. Please extend the discussion and describe in more detail what is seen in

figure 11.

Minor comments and typos: (P refers to the page and L to the line)

P1L3: cut "strongly"

P1L18: "high salinity water" → "saline water"

P2L4: Snow limits also the summer sea ice melt

P2L8-9: Please consider to rewrite this abstract since it is hard to understand

P2L10: "snow melt" → "melted snow"

P2L12: "Uncertainty in knowledge" reads strange, please rephrase

P2L13: "Assessing snow amounts on sea ice is not straight-forward from atmospheric forcing alone" → "Assessing snow amounts on sea ice from atmospheric forcing alone is not straight-forward"

P2L22-23: I think all of these effects can also be observed during winter season.

P2L23-25: Please specify "shallow snow". According to figure 1 the snow depth can be up to 1 m which I would not consider as shallow.

P2L26: "...impact on snow microstructure..." → "impact on the snow microstructure"

P3L2: You could cite a paper dealing with this problem. E.g. Markus et al (2006)

P3L17: Please explain the grain parameters.

P4L15-20: This paragraph is difficult to read. I recommend to rewrite it

P4L26: "in the ocean from below" → "in the sea ice from the ocean below"

P6L13-14: "A critical assumption is. . .. That it assumes..." please rewrite.

P7L13: What does MPFD mean in equation 14?

P7L22: please explain all coefficients in equation 15.

P8L18: "w" → "we"

P8L18: please mention what CFL stands for

P8L25: "and a Dirichlet boundary condition by..." → "and a Dirichlet boundary condition at the bottom by..."

P9L3: Equation 18 needs to be explained in more detailed

P9L20: cut out "for snow"

P10F1: "two sea ice buoys" → "two snow buoys". Please improve the notation of the dates in this figure.

P10L1: In general, the buoys to not measure the snow depth but the hight of the snow above the initial snow/ice interface. In case of the formation of superimposed, this can make a significant difference

P11L17: "after which" → "and"

P11L18: "up to" → "until"

P11L25: to me, a volumetric ice content of 0.9 seems rather low.

P12L7-8: "ERA5... to provide" → ERA5... data to provide"

P14L6-8: This reads strange, please rewrite

P14L7: Please specify "spring" (which months?)

P15L11: Please specify "slightly"

P16L6: "depending" → "depending on"

P17F5: I don't see grey colored snow in figure 5 (b)

P17L9: "For warming" → "For the warming experiment"

P20: Figure 8 is shown before mentioning it in the text

P22L1: Please rewrite the first sentence

P22L25: "simulate sea ice" → "simulated snow on sea ice"

P24: I don't see grey snow in figure 11 (b).

P24F11: "melting conditions is enforced" → "melting conditions are enforced"

P25L3-4: please rewrite

P25L7: "limited forcing data" → "limited measurement data"

P25L7-9: please mention the other limited measurement data

––––––––––––––––––––––––––

---

## Short Comment (SC1) · 26 Jul 2019

Thank you for your efforts to make your code and dat available. There are still some aspects which do not conform to GMD policy and which need to be changed in the revised manuscript.

**Links to SVN repositories**

Links to SVN repositories are not permanent archives of the code. Should the project, for example, change revision control system then these links will go dead and the version numbers will no longer identify the code. Please persistently and publicly archive

the exact version of the code used, and cite the resulting DOI accordingly. Most authors choose to use Zenodo (https://zenodo.org) for this, though any archive compliant with GMD policy will do.

**No run or analysis scripts**

It is at least not clear where the code is that ran the model (model input and configuration files, for examples), nor the code that was used to analyse the results and produce the figures and numerical results in the manuscript. These also form part of the code supporting the manuscript and should be published.

**Data citations are underspecified**

The data from Meereisportal is insufficiently clearly specified (which dataset, which version, which dates etc.) The site itself contains information on how it should be cited, please follow it (http://data.meereisportal.de/gallery/index_new.php?lang=en_ US&survey=&active-tab1=citation&active-tab2=). In particular, URLs can change, so just citing URLs is difficult. Further, it does not seem to be the case that all the data used has been cited. For example, there is mention elsewhere in the manuscript of the use of ERA5 data. Please ensure that all the data and code used in the production of the manuscript is precisely identified and cited from the code and data availability section.

For further details see the GMD code and data availability policy: https://www. geoscientific-model-development.net/about/code_and_data_policy.html

---

## Referee Comment (RC2) · Anonymous Referee #2 · 10 Aug 2019

The analysis is a model demonstration of applying SNOWPACK, a sophisticated snow model, in the Antarctic sea-ice environment. The topic presented here is of great interest to the cryosphere community, and is an excellent example of transferring knowledge from the terrestrial snow environment to the sea-ice environment. The analysis is well-presented, the manuscript being clearly-written and well-organized. There are several assumptions made for the model that need further explanation, especially with regard to how these assumptions compare to the true sea-ice environment and the implications of their differences. Please find comments below that I hope the authors will find useful.

P1, L4. It's the loss of brine during melt that lowers sea-ice salinity, not so much freshwater percolation. Sea ice salinity decreases during the melt season with the

[Figure]

expansion and interconnection of brine channels, leading to melt pond (and melt water) drainage through the ice.

P1, L7. Delete "and" before "to describe water..."

P2, L5-6. And the need to better represent the heterogeneity of these properties?

P2, L8. "the snow cover" do you mean the weight of the snow cover?

P2, L9. "over solely thermal growth" is confusing.

P2, L11-12. This is sentence is redundant with itself. Do you mean that the scales are poorly known because of their limited observations?

P2, L13-15. Sublimation also plays a role.

P2, L22-23. Where is this often observed?

P2, L23. Shallow relative to what?

P3, L1-2. This sentence needs to be a little more specific. There are remote sensing methods of snow that have no reliance on brightness temperatures (e.g. dual laser and radar altimetry).

P3, L13. A sentence could be added here to really set the stage. Something along the lines of:

"In this analysis, we apply SNOWPACK to the Antarctic sea-ice environment and demonstrate its ability to successfully reproduce snow-on-sea ice conditions..." etc.

P3, L22. Typo. "layer's"

P4, L15. Why does the brine freeze instantly? Wouldn't brine salinity increase with decreasing temperature until it reached the eutectic point for salt? Lines 18-19 and 22 and simulation results make me think this process is being accounted for, but this should be clarified in the text here.

P5, L11. For readers unfamiliar with the bucket scheme, a single sentence concisely describing its purpose would be appropriate here.

P9, L21-22. Brine channels become closed off during sea ice growth and thus much of the sea ice below sea level is unsaturated. How might this difference between the model and real-life play into the results?

Figure 1. The dates are not clear at the beginning of the buoy trajectories.

P11, L6. Do you mean here that comparisons between the model simulations and IMB data were made with regard to the sea ice properties only?

P11, L16-17. Please check the dates here. One of them is incorrect.

P11, L23. Typo "was" to "were"

P12, L1-2. Unless the sea ice is melting and drained, it would retain brine content above sea level.

P12, L2-3. Where do these initial values come from, and how sensitive are the results to them?

P12, L11. "measured snow depth" Measured snow depth from the snow buoys is fraught with uncertainties due to blowing and drifting snow, both during and outside of snowfall events. How representative is it to take these measured snow depth increases as snowfall events for the model?

P12, L12. Typo. Change "is below" to "is less than the"

P12, L21. Not all sea ice below sea level is saturated, and not all sea ice above sea level is salt-free. The model assumptions don't show geophysically-realistic vertical profiles of bulk salinity (please see the following for examples: https://apps.dtic.mil/dtic/tr/fulltext/u2/a312027.pdf). It would be worthwhile to discuss the discrepancies with observations, and possible pathways for future improvement of their representation. It would also be worthwhile to discuss the limited representation

of brine salinity – it also does not reflect a geophysically-realistic profile in this first example.

Figure 2 caption "dry sea ice" Technically, this is dry, porous freshwater ice.

P13, L5-6. sea level doesn't stay below the snow-ice interface for the simulation in panel b, does it?

Figure 3. It would be informative to show the temperature difference between panels b and c.

Why are there intermittent decreases in snow depth in panel a compared to what appears to be a steady increase in snow depth in panel b? It would be helpful to increase the y-axis in all of the panels so that readers can see the full range in snow depth. Alternatively, zoomed in panels could be helpful.

In panel a, is the model being re-adjusted to the measured snow depth at each time step or is it melting?

P14, L12. What is meant by an overestimated surface energy balance? If the energy budget is in balance, how can it be overestimated?

P13, L12-13. Do you mean here that snow density was too small, and thus, its insulating capacity was too high, keeping the sea ice "warmer" relative to the observed temperatures?

P15, L7-8. Isn't good agreement expected with this approach?

P15, L11. How much is slightly overestimated? It's cut off in the figure.

P15, L17-18. It would be helpful to state when (which date) flooding occurs since it's not apparent in the figure.

P15, L28-29. Can difference plots be shown? How well do they compare?

Figure 4a. The boundary between the snow and sea ice looks unusual. Is LWC actually

zero in the uppermost layers of the ice surface or is there a gap between the snow and ice in this figure?

Figure 4b. Why does the bulk salinity remain constant both vertically and horizontally in panel b? Why might this differ so much from observations? (please see the following for profile examples: https://apps.dtic.mil/dtic/tr/fulltext/u2/a312027.pdf).

Figure 5 caption, dry snow is colored grey. This description is incorrect here?

Figure 6. This is a nice simulation experiment. For figure 6d, what mechanism is causing the brine salinity to increase in the uppermost part of the snow pack during March 2015? Also, what is happening with brine salinity in the lowermost ice layers from Sept - Dec 2015?

Figure 8d. How can there be brine in the topmost layers of the snow pack?

P25, L9-10. While I agree with this statement, to bolster it, more simulations forced by ERA5 snowfall events rather than the selected events by the IMB data should be conducted and, at the very least, shown in supplementary material. Many current approaches for snow depth simulations are simplified and exclude loss terms. It would be informative to see if this approach is more effective in simulating snow depth due to its sophistication.
* * *

---

## Author Comment (AC1) · 25 Sep 2019

**General response**

We present a thoroughly revised manuscript, addressing the issues raised by the two Reviewers and the Executive Editor. We made the following changes:

- Over the recent months, we have further developed the model, including several bug fixes. For this reason, we redid all simulations, with improved settings and based on the comments below. Particularly the warming example simulation looks different now as a melt pond forms on top of the sea ice. In the first submission, an improper treatment of the upper boundary by the model could lead to the loss of meltwater at the top of the model domain. This has been corrected.

- We revised all plots and added more explanation where appropriate.

- We now provide an outlook section where we address how discrepancies between our simulations and the existing literature are briefly addressed.

- We removed the discussion of the Bucket scheme with fixed, prescribed salinity profiles. We encountered that the implementation was flawed, as the bucket scheme may transport part of the brine downward when it exceeds the size of the "bucket". However, the thermodynamics routine would melt additional sea ice to maintain thermal equilibrium, which in turn again would be transported downward by the water transport scheme. This is undesirable and we were not able to come up with a solution within the time frame of the review process.

Please find a detailed response below, followed by the revised manuscript and a track-changed version of the revised manuscript. Note that page and line numbers refer to the revised manuscript without track changes.

**Response to Reviewers**

**Reviewer #1**

In the manuscript: "Version 1 of a sea ice module for the physics based, detailed, multi-layer SNOWPACK model", an extension of the physical based SNOWPACK model to sea ice is presented. Parts of the model equations are adapted to account for salinity. In addition, other relevant processes like sea ice growth and melt, flooding and brine dynamics are included. The authors discuss the performance of the model comparing it to snow buoy and IMB measurements over Antarctic sea ice. Overall, this manuscript is well written and has a clear structure. Modeling the evolution of snow on sea ice is a challenging task and the here presented model extensions provide important progress for the snow on sea ice modeling community. I believe that this model will be widely used in future research, especially if it will be coupled to SMRT like the authors state in their conclusions. However, the presentation and the discussion of the results are partly too short and important details are missing. Therefore I recommend the manuscript to be published after major revisions. I have three general comments that I would like the authors to address.
*We thank the reviewer for the careful review and constructive comments. Please find our response to the issues raised below.*

**General comments (GC):**

- GC1: In the description of the experiments, important details about the model initialization are missing. In SNOWPACK, the snow is described by grain parameters like dendricity, sphericity and bond radius. These parameters are hard to obtain from snowpit or snow buoy measurements and I wonder what the authors assumed in the model initialization (so far, it is not mentioned in the manuscript). A discussion about these parameters and their influence on the results would be very helpful for further SNOWPACK users. In addition, the authors should provide the full initialization profiles and SNOWPACK settings (I guess this can be provided in the appendix).
  *So far, we used a simple procedure for the initialization of the snow layers, as the buoys were installed with low snow depths, such that the majority of the snow accumulates in the simulations after initialization.*

*We mention this in the revised manuscript P12,L29-31. Otherwise, it is typically recommended to run SNOWPACK for some setup and then select layers that closest match the observed layering in terms of grain size, grain shape and density. Note that the Online supplement contains the input files for the SNOWPACK model as used in this study.*

- GC2: In general, the motivation and discussion of the different experiments (section 4.1 to 4.5) are too short and need to be extended. Some features, which are visible in the figures are not or only poorly discussed. I encourage the authors to carefully read my specific comments for the single chapters.
  *Please find our response to the points raised below. We improved the introduction and discussion of the results throughout the manuscript.*

- GC3: It seems like the authors used the same color-scale range for all figures. In some figures, it is impossible to distinguish between different values since the color-scale goes far beyond the actual values in the figure. I recommend to use individual,representative color-scales for each figure.
  *We revised all the figures.*

**Specific comments:**

- Section 3.1: Please improve figure 1.
  *We improved the readability of the figure.*

- Section 4.1: From figure 2, it looks like it takes less than two hours until the 1.58 m thick sea ice is saturated. Is this realistic? It seems very fast to me.
  *It indeed seems to be an overestimation. We used the arithmetic mean for averaging the hydraulic conductivity between nodes, even though geometric mean is recommended (see (Haverkamp and Vauclin, 1979; Celia et al., 1990)). We therefore repeated the simulations shown in the paper using the geometric mean. We then find a longer time period before saturation is reached (around 16 hours). The saturated hydraulic conductivity for a porosity of $0.06$ is $0.000035474$ m/s. This corresponds to about $13$ cm/hour. We provide now additional explanation in the manuscript (P15,L11-13).*

- The caption for figure 2 (a) is confusing. I guess you mean "dry sea ice" instead of "dry snow"?. In figure 2 (a) to 2 (d) the color-scale range need to be improved.
  *We rephrased the caption and improved the figure (Fig. 4 in the revised manuscript).*

- Section 4.2: Please increase the y-axis in figure 3 (b) Please mention what the blue line in figure 3 (c) is (I guess the ice/ocean interface?).
  *We improved the figure and figure caption (Fig. 5 in the revised manuscript).*

- In figure 3 (d), a difference of 6-10C is found close to the snow/ice interface. Do you expect the differences to be even higher at the top of the snow? As I understood, you use ERA5 incoming longwave radiation as a forcing data. So in principle, the differences shown in figure 3 (d) could also be caused by errors in the forcing data. It might be helpful to add a timeseries of the ERA forcing data to figure 3.
  *We added two figures with the forcing data for the two Snow Buoys (see Figs. 2 and 3 in the revised manuscript). We discuss now that temperature errors can also be a result of errors in the forcing data (P18,L1-2).*

- I'm surprised that even at the ice/ocean interface, the difference can be up to 2C. To my understanding, the temperature should always be around the freezing temperature, or not?
  *The reason is that the simulate ice thickness is slightly underestimated, such that the lower boundary in the simulation is not exactly the lower boundary in the measurements. We explain this now in P16,L5-6.*

- The caption for figure 4 is wrong (there is no temperature plot). Please adapt the color-scale for figure 4 (b).
  *We improved the figure and figure caption (now Fig. 7 in the revised manuscript).*

- Section 4.3: It is not clear to me, what the authors want to demonstrate in this section. I therefore encourage the authors to write a short motivation at the start of the experiments (This also applies for the subsequent

sections). In Figure 5 (a) to 5 (c), vertical stripes are visible at the end of the simulation, which look like numerical instabilities.Are these related to the bucked scheme or to a too big timestep?
*We actually decided to remove the description and simulation of the bucket scheme with fixed, prescribed salinity profiles. We encountered that the implementation was flawed, as the bucket scheme may transport part of the brine downward when it exceeds the size of the "bucket". However, the thermodynamics routine would melt additional sea ice to maintain thermal equilibrium, which in turn again would be transported downward by the water transport scheme. This is undesirable and we were not able to come up with a solution within the time frame of the review process.*

- Section 4.4 In figure 6 (b) and (d), horizontal stripes are visible in the ice in the middle and at the end of the simulation, which look like numerical instabilities. To my understanding, these patterns should not appear since you are using the Crank-Nicolson scheme and the CFL criteria. Do you have an explanation for this phenomena? Please also discuss if these instabilities could have influenced the results of the simulations.
*Crank-Nicolson is not necessarily immune to spurious oscillations when sharp gradients, discontinuities are present (Østerby, 2003). The CFL criterion is not necessary for the Crank-Nicolson scheme, but helps reduce some of the oscillations. The diffusion term helps to dampen the oscillations. There are some other numerical schemes which may exhibit smaller oscillatory behaviour (upwind schemes, Runge-Kutta method), and those could be considered in future revisions of the SNOWPACK model (see P28,L29-31.*

- In figure 6 (d), high brine salinity is simulated at the top of the snowpack. This must be wrong as I can't find any explanation how this could have happened? (especially since the snow was dry according to figure 6 (b)).
*The brine salinity appearing in the near surface layer resulted from numerical rounding errors when calculating bulk salinity, which, given the low temperatures, resulted in very high brine salinities. This rounding error has been resolved now by revising the model source code.*

- Figure 7 (a), (c) and (d) are not discussed in the text.
*We added a discussion of the simulated snow microstructure, see P20,L28-34.*

- Section 4.5. Please extend the discussion of the experiments.
*Please see the revised section (Section 4.4 in the revised manuscript).*

- The color-scale for figure 8 (a) needs to be adjusted.
*We think that the colour bar for 8(a) (Fig. 10(a) in the revised manuscript) spans the temperature range in the simulations, so we are uncertain what the reviewer exactly means here.*

- Section 4.6. Please extend the discussion and describe in more detail what is seen in figure 11.
*We amended the discussion of Fig. 11 (Fig. 13(a) in Section 4.5 in the revised manuscript).*

Minor comments and typos: (P refers to the page and L to the line)

- P1L3: cut "strongly"
*Corrected.*

- P1L18: "high salinity water" → "saline water"
*Corrected.*

- P2L4: Snow limits also the summer sea ice melt
*Amended the text.*

- P2L8-9: Please consider to rewrite this abstract since it is hard to understand
*Rephrased.*

- P2L10: "snow melt" → "melted snow"
*Changed to: "Snow meltwater"*

- P2L12: "Uncertainty in knowledge" reads strange, please rephrase
*Changed to: "limited knowledge"*

- P2L13: "Assessing snow amounts on sea ice is not straight-forward from atmospheric forcing alone" → "Assessing snow amounts on sea ice from atmospheric forcing alone is not straight-forward"
  *Rephrased.*

- P2L22-23: I think all of these effects can also be observed during winter season.
  *It was indeed poorly formulated. We rephrased the sentence.*

- P2L23-25: Please specify "shallow snow". According to figure 1 the snow depth can be up to 1 m which I would not consider as shallow.
  *It was indeed poorly formulated, and we took "shallow" with reference to alpine snowpacks. We rephrased the sentence.*

- P2L26: "...impact on snow microstructure..." → "impact on the snow microstructure"
  *Corrected.*

- P3L2: You could cite a paper dealing with this problem. E.g. Markus et al (2006)
  *Citation added.*

- P3L17: Please explain the grain parameters.
  *Brief explanations and appropriate citation added (P3,L27-32 in the revised manuscript).*

- P4L15-20: This paragraph is difficult to read. I recommend to rewrite it
  *We rephrased this paragraph.*

- P4L26: "in the ocean from below" → "in the sea ice from the ocean below"
  *Rephrased.*

- P6L13-14: "A critical assumption is.... That it assumes..." please rewrite.
  *Rephrased.*

- P7L13: What does MPFD mean in equation 14?
  *We closely follow the notation in Celia et al. (1990), where it is not specified either, but presumably it stands for the right hand side of the Modified-Picard Finite Differences approximation.*

- P7L22: please explain all coefficients in equation 15.
  *Missing coefficients added, including an equation to show the calculation of the fluxes $q$ (see Eq. 15 and 16 in the revised manuscript).*

- P8L18: "w" → "we"
  *Corrected.*

- P8L18: please mention what CFL stands for
  *Amended and citation added.*

- P8L25: "and a Dirichlet boundary condition by..." → "and a Dirichlet boundary condition at the bottom by..."
  *Corrected.*

- P9L3: Equation 18 needs to be explained in more detailed
  *We provide now an additional step in the derivation (see Eq. 20 in the revised manuscript).*

- P9L20: cut out "for snow"
  *Corrected.*

- P10F1: "two sea ice buoys" → "two snow buoys". Please improve the notation of the dates in this figure.
  *Corrected.*

- P10L1: In general, the buoys to not measure the snow depth but the hight of the snow above the initial snow/ice interface. In case of the formation of superimposed, this can make a significant difference
  *Good point that this was poorly formulated. We rephrased this part.*

- P11L17: "after which" $\rightarrow$ "and"
  *Corrected.*

- P11L18: "up to" $\rightarrow$ "until"
  *Corrected.*

- P11L25: to me, a volumetric ice content of 0.9 seems rather low.
  *Indeed, a volumetric ice content of 0.9 is rather low, so we redid the simulations with a volumetric ice content of 0.95, unless stated otherwise.*

- P12L7-8: "ERA5...to provide" $\rightarrow$ "ERA5...data to provide"
  *Corrected.*

- P14L6-8: This reads strange, please rewrite
  *Rephrased.*

- P14L7: Please specify "spring" (which months?)
  *Specified the months.*

- P15L11: Please specify "slightly"
  *This was indeed poorly formulated. Just before our initial submission, we discovered that we interpreted the precipitation rates from the ERA5 data wrongly. After correcting, we didn't adapt the figure scaling and manuscript in accordance with the new simulation result. We apologize for that.*

- P16L6: "depending" $\rightarrow$ "depending on"
  *Corrected.*

- P17F5: I don't see grey colored snow in figure 5 (b)
  *Corrected, it was due to copying figure captions without realizing the sentence was not valid for this figure.*

- P17L9: "For warming" $\rightarrow$ "For the warming experiment"
  *Rephrased to: "To force warming conditions, ..."*

- P20: Figure 8 is shown before mentioning it in the text
  *Corrected.*

- P22L1: Please rewrite the first sentence
  *Rephrased.*

- P22L25: "simulate sea ice" $\rightarrow$ "simulated snow on sea ice"
  *Rephrased it differently. We think that we should stress here that SNOWPACK does not only simulate snow on sea ice, but the snow – sea ice system as a whole.*

- P24: I don't see grey snow in figure 11 (b).
  *Corrected, it was due to copying figure captions without realizing the sentence was not valid for this figure.*

- P24F11: "melting conditions is enforced" $\rightarrow$ "melting conditions are enforced"
  *Corrected.*

- P25L3-4: please rewrite
  *Rephrased.*

- P25L7: "limited forcing data" $\rightarrow$ "limited measurement data"
  *Corrected.*

- P25L7-9: please mention the other limited measurement data
  *This paragraph was poorly formulated and has been revised.*

**Reviewer #2**

The analysis is a model demonstration of applying SNOWPACK, a sophisticated snowmodel, in the Antarctic sea-ice environment. The topic presented here is of great interest to the cryosphere community, and is an excellent example of transferring knowledge from the terrestrial snow environment to the sea-ice environment. The analysis is well-presented, the manuscript being clearly-written and well-organized. There are several assumptions made for the model that need further explanation, especially with regard to how these assumptions compare to the true sea-ice environment and the implications of their differences. Please find comments below that I hope the authors will find useful.
*We thank the reviewer for the insightful and constructive comments. Please find below our response to the issues raised.*

- P1, L4. It's the loss of brine during melt that lowers sea-ice salinity, not so much freshwater percolation. Sea ice salinity decreases during the melt season with the expansion and interconnection of brine channels, leading to melt pond (and melt water) drainage through the ice.
  *We rephrased this sentence to include melt ponds, nevertheless, we found several mentions in literature of flushing of salt out of the sea ice due to freshwater percolation (Notz and Worster, 2009). We now mention melt ponds in the introduction (see P2,L32-35 in the revised manuscript).*

- P1, L7. Delete "and" before "to describe water..."
  *Sentence rephrased in a different way.*

- P2, L5-6. And the need to better represent the heterogeneity of these properties?
  *Good point, rephrased to: "indicating the need to explicitly consider those properties and their spatial and vertical variability."*

- P2, L8. "the snow cover" do you mean the weight of the snow cover?
  *Sentence rephrased "... the snow cover, due to its weight, pushes ...". We think that it is also not correct to say that "the weight of the snow pushes...", as the "weight" cannot act on something.*

- P2, L9. "over solely thermal growth" is confusing.
  *Rephrased.*

- P2, L11-12. This is sentence is redundant with itself. Do you mean that the scales are poorly known because of their limited observations?
  *We meant that the spatial and temporal variability of processes such as flooding and superimposed ice formation are poorly known, because of limited knowledge of snow cover distribution and properties. Rephrased, see P2,L14-15 in the revised manuscript.*

- P2, L13-15. Sublimation also plays a role.
  *Good point! Added.*

- P2, L22-23. Where is this often observed?
  *We found the word "often" too vague and rephrased the sentence, including appropriate references (see P2,L25-28 in the revised manuscript).*

- P2, L23. Shallow relative to what?
  *Similar to our response to a similar comment by Reviewer #1: It was indeed poorly formulated, and we took "shallow" with reference to alpine snowpacks. We rephrased the sentence.*

- P3, L1-2. This sentence needs to be a little more specific. There are remote sensing methods of snow that have no reliance on brightness temperatures (e.g. dual laser and radar altimetry).
  *Good point! Added, with the note that snow properties (depth and density) are still a source of uncertainty in laser or radar altimetry (see P3,L4-7 in the revised manuscript).*

- P3, L13. A sentence could be added here to really set the stage. Something along the lines of: "In this analysis, we apply SNOWPACK to the Antarctic sea-ice environment and demonstrate its ability to successfully reproduce snow-on-sea ice conditions..." etc.
  *Thank you for the suggestion, we inserted a similar sentence (P3,L22-23 in the revised manuscript).*

- P3, L22. Typo. "layer's"
  *Corrected.*

- P4, L15. Why does the brine freeze instantly? Wouldn't brine salinity increase with decreasing temperature until it reached the eutectic point for salt? Lines 18-19 and 22 and simulation results make me think this process is being accounted for, but this should be clarified in the text here.
  *We indeed wanted to express here that with decreasing temperature the brine volume decreases such that the eutectic point for the brine corresponds to the temperature of the surrounding ice or snow. We rephrased the manuscript (see P4,L29-31 in the revised manuscript).*

- P5, L11. For readers unfamiliar with the bucket scheme, a single sentence concisely describing its purpose would be appropriate here.
  *We actually decided to remove the description and simulation of the bucket scheme with fixed, prescribed salinity profiles. We encountered that the implementation was flawed, as the bucket scheme may transport part of the brine downward when it exceeds the size of the "bucket". However, the thermodynamics routine would melt additional sea ice to maintain thermal equilibrium, which in turn again would be transported downward by the water transport scheme. This is undesirable and we were not able to come up with a solution within the time frame of the review process.*

- P9, L21-22. Brine channels become closed off during sea ice growth and thus much of the sea ice below sea level is unsaturated. How might this difference between the model and real-life play into the results?
  *We are not certain if the reviewer means the same with the term saturation. We consider saturated ice as ice where the pore space is completely depleted and the sea ice only consists of freshwater ice and brine. The hydraulic conductivity in our model regulates the flow of brine. So in our model approach, a zero hydraulic conductivity would stop any brine migration through the model domain. This implies, however, that the pressure head and degree of saturation would remain constant in time. However, the initial profile used to set-up the model would also affect the initial and temporal evolution of the brine. We think that the influence of the calculated flooding is relatively small. Pore space in snow is so much larger than in the underlying ice, that very soon ocean water will reach the snowpack. Ultimately, the model we propose is particularly aimed at simulating the snow covering sea ice. Nevertheless, the framework we propose for liquid water flow coupled to the transport equation for salinity may have potential for future development.*

- Figure 1. The dates are not clear at the beginning of the buoy trajectories.
  *We improved the figure.*

- P11, L6. Do you mean here that comparisons between the model simulations and IMB data were made with regard to the sea ice properties only?
  *Yes, due to the way the IMB was installed, we only have the temperature profile in the ice part, and can thus only assess ice temperature and ice thickness. We rephrased the manuscript (see P11,L1-4 in the revised manuscript).*

- P11, L16-17. Please check the dates here. One of them is incorrect.
  *We are not sure what the reviewer meant here, but we rephrased the sentence (see P12,L6-9 in the revised manuscript).*

- P11, L23. Typo "was" to "were"
  *Corrected.*

- P12, L1-2. Unless the sea ice is melting and drained, it would retain brine content above sea level.
  *We mention now that it is a simplification of reality to assign zero brine volume to sea ice above sea level (see P12,L24-27 in the revised manuscript).*

- P12, L2-3. Where do these initial values come from, and how sensitive are the results to them?
  *We provide some additional explanation now in the manuscript. Note that the initial snow covers in our simulations are very small ($10$ cm or less). The majority of the snowpack builds up during the model simulations. See P12,L29-31 in the revised manuscript).*

- P12, L11. "measured snow depth" Measured snow depth from the snow buoys is fraught with uncertainties due to blowing and drifting snow, both during and outside of snowfall events. How representative is it to take these measured snow depth increases as snowfall events for the model?
  *We are not certain where the reviewer aims at. Is it about snow drift disturbing the snow height measurements? First of all, each individual sensor checks for signal consistency and by having four snow height sensors on a buoy, the consistency between the sensors can also be verified. If it is about snow drift removing snow, we would argue that this is precisely why the snow buoy system is advantageous, as it allows for an assessment of the actual amount of snow on sea ice.*

- P12, L12. Typo. Change "is below" to "is less than the"
  *Corrected.*

- P12, L21. Not all sea ice below sea level is saturated, and not all sea ice above sea level is salt-free. The model assumptions don't show geophysically-realistic vertical profiles of bulk salinity (please see the following for examples: `https://apps.dtic.mil/dtic/tr/fulltext/u2/a312027.pdf`). It would be worthwhile to discuss the discrepancies with observations, and possible pathways for future improvement of their representation. It would also be worthwhile to discuss the limited representation of brine salinity – it also does not reflect a geophysically-realistic profile in this first example.
  *The document, which is pointed to here, shows mainly information about the relationship between floe thickness and average bulk salinity. Only one figure shows vertically resolved bulk salinity profiles. The field data shows that typically, bulk salinity decreases with floe thickness, which is often explained due to gravity drainage of dense brine. This is something very challenging to reproduce. We provide now a discussion of our results with possible future model improvements in a new Outlook section (see P28,L24-25 in the revised manuscript).*

- Figure 2 caption "dry sea ice" Technically, this is dry, porous freshwater ice.
  *Thanks for the suggestion. Corrected.*

- P13, L5-6. sea level doesn't stay below the snow-ice interface for the simulation in panel b, does it?
  *Indeed, this was poorly formulated. We misinterpreted ERA5 precipitation rates, and after updating the data set, we forgot to adjust the manuscript accordingly. We rephrased the sentence (see P15,L25-28 in the revised manuscript).*

- Figure 3. It would be informative to show the temperature difference between panels b and c.
  *We created a separate plot for the difference plots, so we can show the difference between a and c, as well as b and c. Please find the new figure 6 in the revised manuscript.*

- Why are there intermittent decreases in snow depth in panel a compared to what appears to be a steady increase in snow depth in panel b? In panel a, is the model being re-adjusted to the measured snow depth at each timestep or is it melting?
  *As we explain now in P14,L14-15 in the revised manuscript, there is no correction in the model when simulated snow depth is above measured snow depth. Decreases in modelled snow depth are due to sublimation and snow melt. The precipitation from ERA5 seems to result in a thicker snowpack on sea ice than found in the snow height driven simulations. We now show cumulative time series of ERA5 precipitation in newly added figures (see Figs. 2 and 3 in the revised manuscript), which show that there is a constant increase in precipitation. This could mask snow depth decreases when there is melt.*

- It would be helpful to increase the y-axis in all of the panels so that readers can see the full range in snow depth. Alternatively, zoomed in panels could be helpful.
  *We improved the figure.*

- P14, L12. What is meant by an overestimated surface energy balance? If the energy budget is in balance, how can it be overestimated?
  *It was indeed poorly phrased, but we meant that the incoming energy could be overestimated, or the outgoing energy be underestimated. We rephrased the sentence.*

- P13, L12-13. Do you mean here that snow density was too small, and thus, its insulating capacity was too high, keeping the sea ice "warmer" relative to the observed temperatures?
  *We assume this refers to P14. It's indeed what was meant here. We rephrased this sentence to be more clear.*

- P15, L7-8. Isn't good agreement expected with this approach?
  *Written as it was, it was indeed a trivial statement. We rephrased the paragraph.*

- P15, L11. How much is slightly overestimated? It's cut off in the figure.
  *We misinterpreted ERA5 precipitation rates, and after updating the data set, we forgot to recheck the figures and text in the manuscript. We updated the figure and rephrased the text.*

- P15, L17-18. It would be helpful to state when (which date) flooding occurs since it's not apparent in the figure.
  *Actually, there is no flooding occurring. We meant to say here that the added weight of snow is increasing the bottom pressure, leading to an influx of saline water. The freeboard remains positive, i.e., the sea level stays below the snow–ice interface. We rephrased the sentence (see P18,L31-32 in the revised manuscript).*

- P15, L28-29. Can difference plots be shown? How well do they compare?
  *We removed the simulation using the bucket scheme and fixed salinity profiles, as we found there were some discrepancies in this model approach, which could not be easily resolved.*

- Figure 4a. The boundary between the snow and sea ice looks unusual. Is LWC actually zero in the uppermost layers of the ice surface or is there a gap between the snow and ice in this figure?
  *There is no gap in the snow and ice in the figure. There is low liquid water content just above the sea level due to capillary suction. We now explicitly mention this: see P18,L26-27 in the revised manuscript.*

- Figure 4b. Why does the bulk salinity remain constant both vertically and horizontally in panel b? Why might this differ so much from observations? (please see the following for profile examples: `https://apps.dtic.mil/dtic/tr/fulltext/u2/a312027.pdf`).
  *When we initialize the simulations with a constant brine salinity profile, the only way the simulations would show a temporally varying bulk salinity is when there is vertical flow of brine. In the simulations this typically happens upon new snow accumulations adding weight. In reality there is also gravity drainage of dense brine, which is not reproduced by our model. We discuss this now in an outlook section at the end of the manuscript (see P28 in the revised manuscript).*

- Figure 5 caption, dry snow is colored grey. This description is incorrect here?
  *Corrected, it was due to copying figure captions without realizing the sentence was not valid for this figure.*

- Figure 6. This is a nice simulation experiment.For figure 6d, what mechanism is causing the brine salinity to increase in the uppermost part of the snow pack during March 2015?
  *The brine salinity appearing in the near surface layer resulted from numerical rounding errors when calculating bulk salinity, which, given the low temperatures, resulted in very high brine salinities. This rounding error has been resolved now by revising the model source code.*

- Also, what is happening with brine salinity in the lower most ice layers from Sept - Dec 2015?
  *Issue resolved.*

- Figure 8d. How can there be brine in the topmost layers of the snow pack?
  *Similar to the previous issue: The brine salinity appearing in the near surface layer resulted from numerical rounding errors when calculating bulk salinity, which, given the low temperatures, resulted in very high brine salinities. This rounding error has been resolved now by revising the model source code.*

- P25, L9-10. While I agree with this statement, to bolster it, more simulations forced by ERA5 snowfall events rather than the selected events by the IMB data should be conducted and, at the very least, shown in supplementary material. Many current approaches for snow depth simulations are simplified and exclude loss terms. It would be informative to see if this approach is more effective in simulating snow depth due to

its sophistication.

*We rephrased this paragraph. We plan on extending the analysis for the other snow buoys, but we feel that it would tend to be out of the scope for a model description paper.*

**David Ham, Executive Editor**

Thank you for your efforts to make your code and dat available. There are still some aspects which do not conform to GMD policy and which need to be changed in the revised manuscript.
*Thank you for the comments. Please find our responses below.*

**Links to SVN repositories**

Links to SVN repositories are not permanent archives of the code. Should the project,for example, change revision control system then these links will go dead and the version numbers will no longer identify the code. Please persistently and publicly archive the exact version of the code used, and cite the resulting DOI accordingly. Most authors choose to use Zenodo (`https://zenodo.org`) for this, though any archive compliant with GMD policy will do.
*We understand the need for a unique identification of the source code. However, we decided against uploading the source code to Zenodo and acquiring a unique DOI, because we always encourage users to use the latest version of the model. We therefore decided to include the source code in the Online Supplement, together with the run files and scripts.*

**No run or analysis scripts**

It is at least not clear where the code is that ran the model (model input and configuration files, for examples), nor the code that was used to analyse the results and produce the figures and numerical results in the manuscript. These also form part of the code supporting the manuscript and should be published.
*Please note that we included all the model input and configuration files in the Online Supplement to the manuscript. The README.txt file contained for each figure in the manuscript the appropriate command to execute SNOW-PACK. We now added the postprocessing and plotting scripts as well.*

**Data citations are underspecified**

The data from Meereisportal is insufficiently clearly specified (which dataset, which version, which dates etc.) The site itself contains information on how it should be cited, please follow it (`http://data.meereisportal.de/gallery/index_new.php?lang=en_US&survey=&active-tab1=citation&active-tab2=`). In particular, URLs can change, so just citing URLs is difficult. Further, it does not seem to be the case that all the data used has been cited. For example, there is mention elsewhere in the manuscript of the use of ERA5 data. Please ensure that all the data and code used in the production of the manuscript is precisely identified and cited from the code and data availability section. For further details see the GMD code and data availability policy: `https://www.geoscientific-model-development.net/about/code_and_data_policy.html`
*We now refer the Snow Buoys with the specific DOIs. Thermistor Buoy 2016T41 does not have a unique identifier or version number.*

**References**

[revised manuscript text omitted]

---

## Referee Report (RR1)

2nd Review for:  Version 1 of a sea ice module for the physics based, detailed, multi-layer SNOWPACK model

I thank the authors for their careful consideration of the reviewers comments. Overall, the revised manuscript is a clear improvement and the new figures provide important details that were missing in the first version. I'm satisfied with the authors response to my comments and have only one minor open question:

**Specific comments:**

- P20L14-17: In this paragraph it is worth to mention the density with which the model initializes  new snow.

**Minor comments and typos:**

- P2L7: The thermal conductivity of snow also varies → remove "also"
- P2L30: Inhomogeneities caused by  → Inhomogeneities in the snow caused by
- P2L33: up to a few decimeter for ... → up to a few decimeter thick for…
- P5L11: ...on local temperature would reduce the energy → remove "would"
- P11L12: ...no diffusion of salt with the atmosphere → "with" seems not to be  the right word
- P15L8: Please add a citation of the ERA5 data
- P16L12: decrease in snow depth → decrease in measured snow depth
- P23L8: Fig. 9b shows that much of.. → Fig. 9b shows that a substantial amount of
- P23L10: $0^{circ}C$ → $0°C$
- P23L12: winter, the flooding → remove "the"
- P24L3: shown in 9d → shown in figure 9d
- P27L5: To test how our model simulations would→ To test how the model would
- P27L21: ,such that that the → remove one "that"
- P32L1:  2 cm of ice, with constant… → 2 cm of ice and constant
-

---

## Author Response (AR2)

**Author response**

We thank the reviewer for the comments. We modified the manuscript accordingly, as detailed below. Proof-reading by all the authors also lead to some additional corrections and modifications, as highlighted in the track-changed manuscript following the response to the reviewer's comments.

- P20L14-17: In this paragraph it is worth to mention the density with which the model initializes new snow.
  *New snow density is determined following a parameterization which takes into account several effects, such as wind, temperature and relative humidity. We provide now additional citation of the parameterization used, and briefly discuss the impact it may have on the simulations (see P18L15-19 in the tracked-changed manuscript in the following pages).*

Minor comments and typos:

- P2L7: The thermal conductivity of snow also varies → remove "also"
  *Corrected.*

- P2L30: Inhomogeneities caused by → Inhomogeneities in the snow caused by
  *Corrected.*

- P2L33: up to a few decimeter for ... → up to a few decimeter thick for...
  *Corrected.*

- P5L11: ...on local temperature would reduce the energy → remove "would"
  *Corrected.*

- P11L12: ...no diffusion of salt with the atmosphere → "with" seems not to be the right word
  *Replaced by "into".*

- P15L8: Please add a citation of the ERA5 data
  *We added the citation.*

- P16L12: decrease in snow depth → decrease in measured snow depth
  *Corrected.*

- P23L8: Fig. 9b shows that much of.. → Fig. 9b shows that a substantial amount of
  *Corrected.*

- P23L10: 0 circ C → 0°C
  *Corrected.*

- P23L12: winter, the flooding → remove "the"
  *Corrected.*

- P24L3: shown in 9d → shown in figure 9d
  *Corrected.*

- P27L5: To test how our model simulations would → To test how the model would
  *Rephrased as: "To test how the model reacts to".*

- P27L21: ,such that that the → remove one "that"
  *Corrected.*

- P32L1: 2 cm of ice, with constant... → 2 cm of ice and constant
  *Corrected.*

[revised manuscript text omitted]